# Phylogenomics reveals the history of host use in mosquitoes

John Soghigian [1,2], Charles Sither[1], Silvia Andrade Justi[3,4,5], Gen Morinaga[2], Brian K. Cassel[1], Christopher J. Vitek[6], Todd Livdahl [7], Siyang Xia[8], Andrea Gloria-Soria[8,9], Jeffrey R. Powell[8], Thomas Zavortink[10], Christopher M. Hardy[11], Nathan D. Burkett-Cadena [12], Lawrence E. Reeves [12], Richard C. Wilkerson [3], Robert R. Dunn[13], David K. Yeates [14], Maria Anice Sallum [15], Brian D. Byrd[16], Michelle D. Trautwein[17], Yvonne-Marie Linton[3,4,5], Michael H. Reiskind[1] & Brian M. Wiegmann [1] ✉

Mosquitoes have profoundly affected human history and continue to threaten human health through the transmission of a diverse array of pathogens. The phylogeny of mosquitoes has remained poorly characterized due to difficulty in taxonomic sampling and limited availability of genomic data beyond the most important vector species. Here, we used phylogenomic analysis of 709 single copy ortholog groups from 256 mosquito species to produce a strongly supported phylogeny that resolves the position of the major disease vector species and the major mosquito lineages. Our analyses support an origin of mosquitoes in the early Triassic (217 MYA [highest posterior density region: 188–250 MYA]), considerably older than previous estimates. Moreover, we utilize an extensive database of host associations for mosquitoes to show that mosquitoes have shifted to feeding upon the blood of mammals numerous times, and that mosquito diversification and host-use patterns within major lineages appear to coincide in earth history both with major continental drift events and with the diversification of vertebrate classes.

Mosquitoes profoundly affect humans, primarily through their ability to transmit pathogenic viruses, nematodes, and protozoa[1]. Each year, their bites transmit millions of human infections, resulting in over 400,000 deaths worldwide[1]. Historically, the toll from mosquito-borne pathogens has been even greater, with consequences for human evolution. For example, the selective pressure posed by human malaria, which is transmitted exclusively by *Anopheles* mosquitoes, has led to human adaptations that lessen the impact of infections such as

[1]Department of Entomology and Plant Pathology, North Carolina State University, Raleigh, NC, USA. [2]Faculty of Veterinary Medicine, University of Calgary, Calgary, AB, Canada. [3]Walter Reed Biosystematics Unit, Smithsonian Institution Museum Support Center, Suitland, MD, USA. [4]One Health Branch, Walter Reed Army Institute of Research, Silver Spring, MD, USA. [5]Department of Entomology, Smithsonian Institution National Museum of Natural History, Washington, DC, USA. [6]Center for Vector-Borne Diseases, University of Texas Rio Grande Valley, Edinburg, TX, USA. [7]Department of Biology, Clark University, Worcester, MA, USA. [8]Department of Ecology and Evolutionary Biology, Yale University, New Haven, CT, USA. [9]Department of Entomology, Center for Vector Biology & Zoonotic Diseases, The Connecticut Agricultural Experiment Station, New Haven, CT, USA. [10]Bohart Museum of Entomology, University of California, Davis, CA, USA. [11]CSIRO Land and Water, Canberra, ACT, Australia. [12]Florida Medical Entomology Laboratory, Institute of Food and Agricultural Sciences, University of Florida, Vero Beach, FL, USA. [13]Department of Applied Ecology, North Carolina State University, Raleigh, NC, USA. [14]Australian National Insect Collection, CSIRO National Collections and Marine Infrastructure, Canberra, ACT, Australia. [15]Departamento de Epidemiologia, Faculdade de Saude Publica, Universidade de Sao Paulo, Sao Paulo, Brazil. [16]College of Health and Human Sciences, School of Health Sciences, Western Carolina University, Cullowhee, NC, USA. [17]Entomology Department, Institute for Biodiversity Science and Sustainability, California Academy of Sciences, San Francisco, CA, USA. ✉e-mail: bwiegman@ncsu.edu

sickle cell, Duffy factor, thalassemia, and glucose-6-diphosphatase deficiency, and these traits are maintained in areas where malaria is common[2,3]. Likewise, viral illnesses such as yellow fever are obligately mosquito-transmitted and have repeatedly shaped the course of human history[4]. And yet, we know little about the long evolutionary history of their family—the Culicidae—nor how so many species of these insects came to be our enduring enemies.

Perhaps surprisingly, given the medical importance of certain mosquito genera, relationships between major groups remain largely unknown. The most comprehensive phylogenetic analyses have focused mostly on specific species-groups or lineages (e.g., the Aegypti Group[5] or the *Anopheles gambiae* complex[6,7]), with relatively little evaluation of relationships above the taxonomic level of genus. The few studies using molecular approaches to evaluate older phylogenetic relationships among the Culicidae are based on relatively limited data[8–11] and failed to definitively resolve the relationships among the most ancient lineages of Culicidae. Thus, the understanding of mosquito phylogeny has remained largely unchanged since the classic morphological analyses of the mid-20th century[12]. The taxonomies built on these putative relationships remain contested, as best exemplified by the numerous nomenclatural changes to the genus *Aedes* in the late 20th and early 21st century[13–15].

Because of large uncertainties in mosquito phylogeny, it has proven challenging to understand the evolution of mosquito traits, including the propensity to feed on humans, how they transmit particular kinds of pathogens, become invasive, or serve as new vectors for emerging pathogens. Out of roughly 3600 mosquito species globally, ~100 species from eleven genera certainly play a role in the transmission of disease to humans[1] and another 200 are likely or potential vector species[16]. It is not yet known how many origins of human-feeding those 100-some species represent, or how specific feeding associations may have influenced diversification in mosquitoes. Evolutionary transitions in broader feeding and habitat preferences, mating interactions, and vagility of species have contributed to the current phylogenetic diversity of many groups of organisms[17–19], and yet relatively little is known about how these factors have shaped the contemporary species richness of mosquitoes. A renewed understanding of mosquito phylogeny provides an explicit evolutionary context for interpreting the major transitions in mosquito feeding habits and a narrative for how these may have been influenced by the environments they inhabit and the hosts they prey upon.

Ground-breaking comparative genomics research in mosquitoes has catalyzed efforts to better understand ecological, behavioral, and morphological adaptations that facilitate their success as human disease vectors[6,20,21], but these studies have largely been limited to species or lineages of medical importance (<5% of all Culicidae). Large-scale phylogenomics has transformed systematic analyses by enabling resolution of relationships among phylogenetically diverse lineages with unprecedented accuracy thanks to much larger data sets[22,23], but these methods have, until now, not been used in mosquitoes. Here, we used a probe-based anchored hybrid enrichment method (AHE)[24] to obtain and sequence hundreds of orthologous genes from fresh and museum specimens and combine these data with existing mosquito genomic resources. Our analysis draws on 53 published genomes and transcriptomes, as well as on newly sequenced AHE data from an additional 215 mosquito species. We present the phylogenomic analysis of the entirety of the Culicidae, proposing a well-supported phylogenetic tree of the family, upon which we examine the evolutionary history of their blood host associations, a critical adaptation that makes these insects so injurious to humans and livestock. Our analyses resolve the diversification of major mosquito lineages, and reveal that mosquitoes originated in the early Triassic (217 MYA [highest posterior density region, HPD: 188–250 MYA]). Then, we use an extensive database of host associations and find support for an ancient, amphibian-feeding ancestor of the Culicidae, with more recent transitions to mammals and birds following the Cretaceous-Paleogene extinction event.

## Results

This phylogenomic dataset is the largest yet assembled for phylogenetically studying mosquitoes, encompasses samples from six continents, with species from both currently recognized subfamilies (Culicinae and Anophelinae), 24 genera, and nine tribes (Supplementary Note 1). From these samples, we recovered the orthologous nucleotide sequences of 709 single-copy genes in more than 203 (>75%) of the species sampled (Supplementary Note 1). Our taxon sampling (number of species) more than doubles previous phylogenetic studies of mosquitoes and samples 40 times as many loci as these studies did. To account for rapid phylogenetic signal decay at the nucleotide sequence level in certain codon positions (saturation – Supplementary Note 2), we analyzed multiple sequence alignments of amino acids and the second nucleotide position in codons, respectively. In total, our alignment contains 523,035 amino acid positions. We inferred phylogenies in IQTree2[25] using maximum likelihood for the nucleotide position two and amino acid alignments, respectively, as well as with ASTRAL based on gene trees using maximum likelihood on individual amino acid alignments of genes. Topologies were highly congruent, with the only differences found within subgenera (Supplementary Figs. 1, 2, 9–13). These phylogenies provide detailed insight into the phylogenetic relationships of major mosquito lineages.

We find strong support for the monophyly of both currently recognized mosquito subfamilies, and all sampled tribes in the subfamily Culicinae (Fig. 1). Within the subfamily Culicinae, the species-poor tribe Aedeomyiini (a pantropical tribe of only seven species) is the earliest diverging tribe, followed by the Uranotaeniini. Interestingly, we recover the enigmatic tribe Toxorhynchitini, which contains the genus *Toxorhynchites*, with carnivorous larvae, as sister to the Sabethini (Fig. 1A). The affinities of *Toxorhynchites* to other mosquitoes have long been uncertain, and until the early 2000s[26] the genus was placed in a monotypic subfamily due to its unique morphology and behavior. Our results provide strong evidence that these non-biting, ornate mosquitoes originated within the Culicinae, as sister to another ornate group of mosquitoes, the Sabethini (Fig. 1A). This finding is particularly intriguing as the Toxorhynchitini (in *Toxorhynchites*) and the Sabethini (in *Malaya* and *Topomyia*) contain the only three exclusively non-biting genera of mosquitoes[27].

In our analyses, all but two genera are monophyletic, consistent with recent analyses that analyzed less data[10]. Those two genera are the species-rich, medically important groups *Aedes* (species of which transmit many pathogenic viruses such as Zika, dengue, chikungunya, and yellow fever)[28] and *Culex* (species of which transmit viruses that cause West Nile and Japanese encephalitis)[28]. It is clear that these clades have complex histories that will continue to require taxonomic study and clarification, but this highlights that overall, the morphological systematics of mosquitoes has performed quite well at delineating clades, if not always in determining the associations between them (see our Supplementary Discussion on mosquito systematics in light of our results).

Our phylogenomic analyses and Bayesian divergence time estimations[29] (Fig. 2, Supplementary Fig. 3) push back the phylogenetic age of mosquitoes considerably[22] and strongly support the hypothesis that mosquitoes originated in the Triassic (~217 MYA [highest posterior density region, or HPD: 188–250 MYA]), before the major radiations of both dinosaurs and mammals, when the dominant vertebrates were crocodile-like reptiles, called archosaurs. This age precedes that of any existing mosquito fossil, the oldest being *Priscoculex burmanicus* from the Early Upper Cretaceous[30]. Our analyses indicate that the last common ancestors of the two extant subfamilies of Culicidae, the Culicinae and the Anophelinae, lived during the early Jurassic, near the Toarcian Warm Interval (~179 MYA [HPD:147–213 MYA] (Fig. 1B),

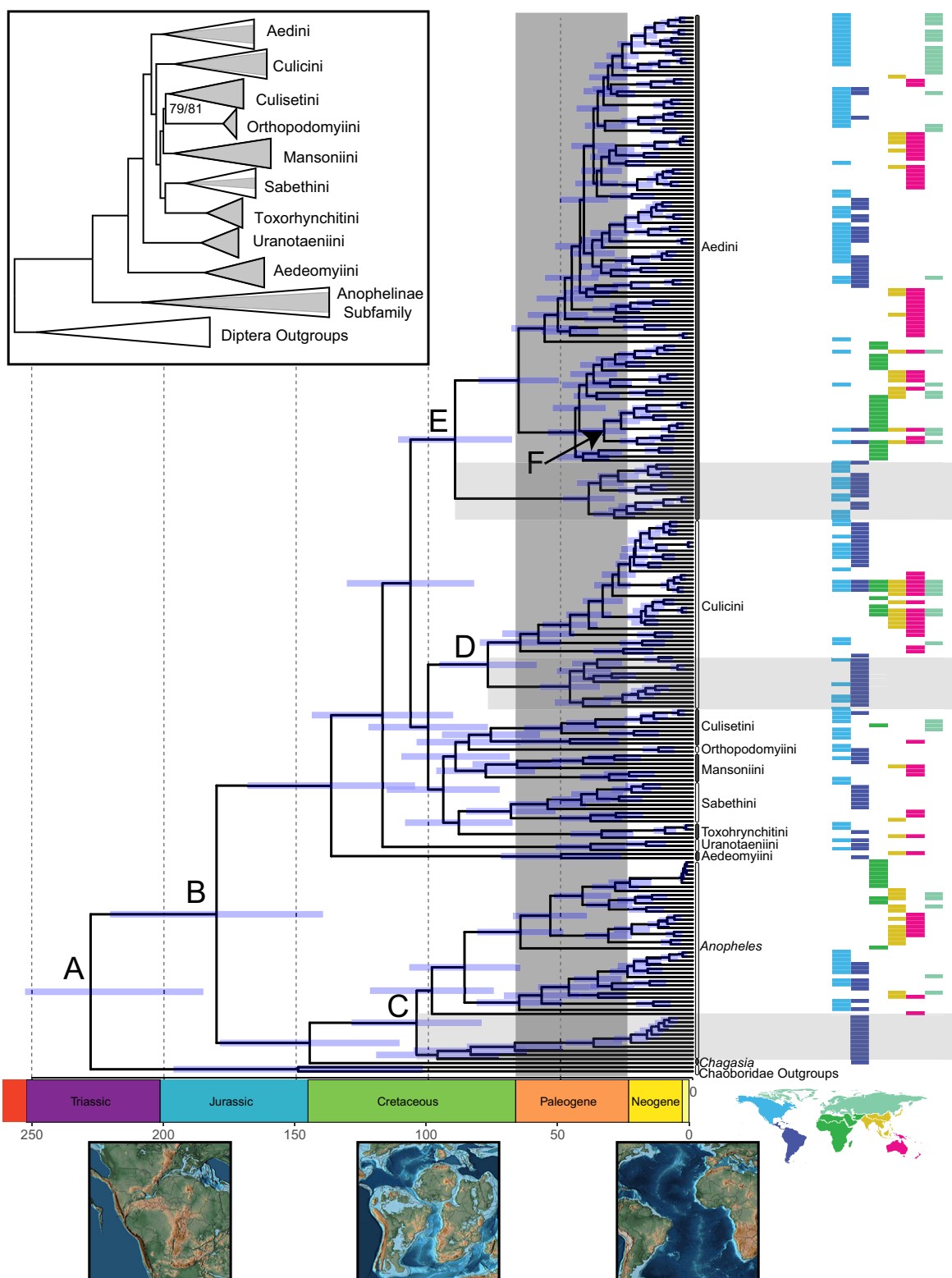

**Fig. 1 | Phylogenetic relationships among major lineages of mosquitoes (Culicidae), as inferred by maximum likelihood and dated using a fossil-calibrated relaxed clock analysis.** The analysis is based on the analysis of amino acid sequences of 709 genes (525,000 aligned sites) from 256 species and calibrated at seven time points[30,70,94–98]. Two Chaoborid outgroups were used to root the phylogeny. Horizontal gray boxes indicate major clades confined to the Americas, whose divergence times correlate with the dates of major geological events. Bars on nodes are 95% HPDs. Paleogeographic reconstructions are from PALEOMAP[99], and colored bars correspond to the biogeographic region of the species listed, as indicated in the world map below. In the inset, tribes have been collapsed for easier viewing, and the proportion of genera sampled are indicated by gray triangles. Support values are SH-like aLRT and Ultrafast Bootstrap values and are only shown if branch support was below 99. **A** The common ancestor of the Chaoboridae and the Culicidae. **B** The common ancestor of all extant Culicidae. **C** The common ancestor of all *Anopheles* in our analysis. **D** The common ancestor of all *Culex* in our analysis. **E** The common ancestor of the tribe Aedini. **F** The common ancestor of all *Aedes (Stegomyia)* sampled in our analysis. Lettered annotations are discussed in greater detail in the text.

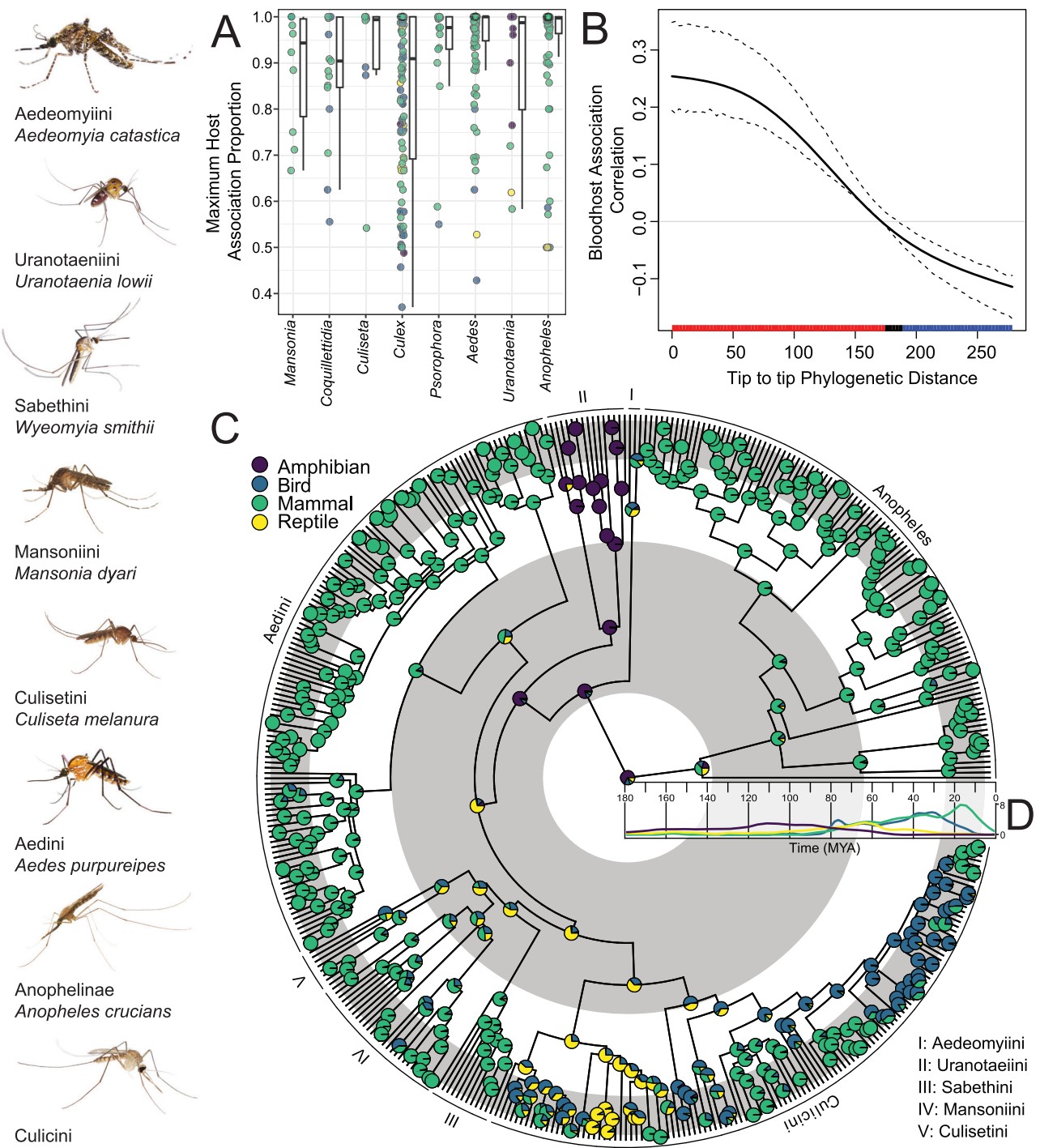

Aedeomyiini
*Aedeomyia catastica*

Uranotaeniini
*Uranotaenia lowii*

Sabethini
*Wyeomyia smithii*

Mansoniini
*Mansonia dyari*

Culisetini
*Culiseta melanura*

Aedini
*Aedes purpureipes*

Anophelinae
*Anopheles crucians*

Culicini
*Culex biscaynensis*

I: Aedeomyiini
II: Uranotaeiini
III: Sabethini
IV: Mansoniini
V: Culisetini

**Fig. 2 | Macroevolutionary analyses of host associations for blood-feeding in mosquitoes based on 293,308 bloodmeal records from 422 different mosquito species. A** The maximum host association score among several genera indicating that many genera have a high degree of class-level host association, although there are exceptions. Each point is a species in a given genus. Boxplots for each genus show median max host association values as black bars, along with interquartile ranges as boxes. Whiskers are drawn to +/−1.5 times the interquartile range. **B** Vertebrate host association contains significant phylogenetic signal as estimated by a phylogenetic Mantel test (shown) and multivariate Blomberg's $K = 0.06$, $P < 0.027$. Among the Culicinae, closely related species have more similar blood-hosts, as indicated by the red bar (significance at an alpha value of 0.05), while more

distant lineages differ in blood-host, as indicated by the blue bar. Dashed lines are bootstrapped (1000 replicates) 95% confidence intervals around the estimate of phylogenetic signal (shown as a solid line). **C** Ancestral state reconstruction on blood-host association in the Culicidae. Our models suggest an amphibian feeding ancestor for all Culicidae. Gray and white bands indicate geologic time periods. **D** Origin and diversification of extant family-level lineages from major vertebrate classes reconstructed from mean lineage through time estimates from VertLife.org posterior phylogenetic estimates. Many major extant lineages of reptiles and amphibians originated during the Jurassic and Cretaceous[92,93], while most modern mammals and birds originated in the late Cretaceous and Paleogene[33,91].

100 MY older than suggested by Misof et al.[22] in their seminal phylogenomic study on the evolutionary history of insects. Our greater sampling of the Culicidae and our integration of multiple mosquito fossils, along with analytical differences (see Methods and Supplementary Discussion), likely account for the differences in our estimates and theirs, as Misof et al.[22] used only one fossil calibration across all flies and had only two mosquito species in their analysis.

The subfamily Culicinae originated in the Cretaceous and diversified therein, with all extant tribes appearing before the Paleogene. The earliest diverging tribes of the Culicinae, Aedeomyiini (133 MYA [HPD:109–162 MYA]) and Uranotaeniini (114 MYA [HPD:94–138]), have contemporary distributions consistent with lineages originating on Gondwana (Fig. 1). A rapid radiation in the late Cretaceous, from 104 [HPD: 87-127 MYA] to 87 MYA [HPD:71-106 MYA], gave rise to the remaining tribes. This time coincides with that of the diversification of flowering plants[31], which adult mosquitoes use for nectar and whose water-filled parts are inhabited by the immature stages of many culicine tribes.

## Discussion

We find a compelling association between mosquito lineages and major geologic events, such that the breakup of Gondwana during the Cretaceous may have shaped the distribution of extant mosquito lineages. These resulting ancient divergences continue to affect modern patterns of mosquito species diversity and distribution, as well as modern patterns in the distribution of mosquito-vectored pathogens and their deadly consequences. For instance, in the subfamily Anophelinae, the lineages confined to what is now the Americas diverged first in the form of the genus *Chagasia* during the early Cretaceous, 142 MYA [HPD:116–172 MYA] (Fig. 1). Approximately 40 million years later, the separation of Gondwana is reflected in the divergence times of *Anopheles* lineages: the *Anopheles* subgenera associated exclusively with the Americas diverged from all other *Anopheles* 103 MYA [HPD: 83–126 MYA] (Fig. 1C). This date coincides with the estimated time of formation of the channel separating South America and Africa in the equatorial zone[32], an event that could have trapped the ancient ancestors of American anopheline malaria vectors in what is now South America. All other *Anopheles* subgenera belong to a second clade of *Anopheles*, which was free to exploit other continents, and as such, have contemporary, cosmopolitan distributions quite unlike their American counterparts. The diversification of *Anopheles* highlights how continental drift events during the Cretaceous may have shaped the present-day diversity of critical disease vectors.

Multiple lineages in the Culicinae reflect the patterns seen in the Anophelinae. Among the Culicini, divergence times and contemporary distributions indicate ancient geographical isolation of lineages corresponding to continental drift events. Two major *Culex* clades, one exclusively found in the Americas and the other now cosmopolitan outside the Americas, diverged 77 MYA [HPD:64–95 MYA] (Fig. 2D). This divergence, if correlated with continental drift events as it appears, thus happened after the deepening of the central and southern South Atlantic that occurred 85–100 MYA[32] as Gondwana split.

In the tribe Aedini, the genus *Psorophora* diverged 87 MYA [HPD:70–106] (Fig. 1E); this genus is confined to the Americas, consistent with the possibility that continental drift events have influenced the diversity of many present-day lineages. The remaining genera in the Aedini share a common ancestor 62 MYA [HPD: 52–77 MYA], with two major clades composed of *Aedes* species as well as other aedine genera (Fig. 1, Fig. S1, Fig. S3). Although the two lineages of *Aedes* originated near the Cretaceous–Paleogene extinction event (K-Pg), a rapid lineage diversification in both clades took place 40–50 MYA. This pattern mirrors recent results that identified a post K-Pg delay in mammal lineage diversification[33], the primary vertebrate hosts for a large majority of *Aedes* species.

Remarkably, many mosquito subgenera are tens of millions of years old. For instance, the two most important disease vectors in the *Aedes* subgenus *Stegomyia* that have become closely associated with humans in behavior, habitat preferences, and invasiveness, *Aedes* (*Stegomyia*) *aegypti* and *Aedes* (*Stegomyia*) *albopictus*, diverged from one another 31 MYA [HPD:25–38 MYA] (Fig. 1F). Indeed, this ancient divergence indicates that although these species are competent vectors for many of the same flaviviruses, this is not due to ancient coevolution between vectors and hosts, given the comparatively recent divergences of flaviviruses[34].

Understanding the phylogenetic relationships among mosquito species and higher taxa is a first step toward understanding the evolutionary basis of key mosquito traits, such as blood-feeding, host choice, and ability to transmit pathogens. To date, evaluation of how evolutionary relationships among disease vectors influence vectorial ability or phenotypic traits have been largely limited to evaluations of subgenera or closely related genera[10,35]. We linked insights from our phylogenomic analysis with data on contemporary blood host information to study the evolution of host associations through time, providing an example of how the availability of a comprehensive mosquito phylogeny allows for insights into important mosquito phenotypes.

Mosquito-host associations are determined by collecting engorged females from the field and subsequently identifying the blood-meal source using a range of techniques from ELISA to PCR. We compiled a comprehensive database of 293,308 blood meal records from 422 different species of mosquito based on the published literature, available on the Culicitree GitHub (https://github.com/jsoghigian/culicitree) and DataDryad. This was possible thanks to the meticulous, mosquito-by-mosquito research of generations of biologists from whose 142 peer-reviewed publications these records were gathered. In total, 391 species had at least two blood-meal records, 326 had more than 5 records and 210 had more than 50 records. For each species of mosquito, we considered the proportion of blood meal records per taxonomic class of hosts to be a measure of the species host associations, and placed unsampled species into our dated phylogeny using a birth-death process model and current taxonomy[36].

Mosquitoes take blood meals predominantly from terrestrial vertebrate hosts (amphibians, birds, mammals, reptiles), though at least one species is known to be specialized on fish[37] and another on invertebrate (annelid) hosts[38]. Individual mosquito species have an association with a particular class of host (Fig. 2A, Supplementary Figs. 4, 5, mean maximum host $a = 0.89$, range = 0.37 to 1), almost certainly due to differences in the traits necessary to locate a host, to overcome host defensive behaviors and immune responses, and to successfully penetrate tiny blood vessels of particular hosts or digest specific blood components. For instance, nearly all (96%) of the blood meals of the malaria vector *Anopheles gambiae* were derived from mammals (27,267), with only occasional blood meals from birds (107 blood meals−0.37%), and a pair of unlucky amphibians (2−0.07%). These individual-level host associations in some cases extend to genera−for instance, *Uranotaenia* had a high association with amphibians, whereas *Aedes* and *Anopheles* display a high association with mammals (Fig. 2A, Supplementary Fig. 4, Supplementary Figs. 5, 15). Other genera−particularly *Culex*, but also *Coquillettidia* and others−are broad generalists and feed on two or more host classes (though even within these genera specialization can exist within species, subgenera, and other taxa). Future studies may reveal which morphological and physiological traits of genera correlate with narrow (e.g., *Uranotaenia*) or broad (e.g., *Culex*) host associations, with important implications for human health. For example, the wide host breadth of many species of *Culex* subgenus *Culex* mosquitoes partially explains the roles these mosquitoes play as vectors of emerging zoonoses. It is such feeding behavior that allows the species *Culex pipiens*, to be a vector of West Nile from birds to humans. Likewise, the predominance of mammal

feeding in *Anopheles* and *Aedes* may have predisposed species within these genera to successfully adapt to not only human feeding, but also to our domesticated livestock and pets, and thus facilitated an association with human settlements[21,39].

Beyond individual differences between species, evolutionary relationships among Culicidae are a significant predictor of host associations (multivariate Blomberg's $K = 0.06$, $P < 0.027$; Fig. 2B, Supplementary Figs. 16–20). Tribes differ substantially in host associations, likely reflecting niche partitioning by host taxon at deeper evolutionary time periods corresponding to the formation of tribes observed in the present, with canalized adaptation to blood-host class within mosquito genera/subgenera in recent time.

While our estimates of divergence times in mosquitoes highlights how ancient mosquito lineages may reflect continental drift events, our ancestral state reconstructions demonstrate how ancient and contemporary mosquito lineages have been shaped by the evolution of vertebrate hosts (Fig. 2D, Supplementary Fig. 7). An amphibian feeding ancestor of the Culicidae is supported across a range of reconstructions (Supplementary Discussion, Supplementary Figs. 21–23), demonstrating the robustness of this finding (Fig. 2C). This is concordant with our inferred timing of the origin of mosquitoes in the Triassic. Fossil and molecular evidence suggests that Gondwanan wetland habitats, the likely ancestral habitat of mosquitoes, would have had ample amphibian blood hosts at the origin of the family, 217 MYA[40]. Interestingly, the nearest blood-feeding relative of mosquitoes, Corethrellidae, feed exclusively on amphibians[41].

The reconstructions within the Culicinae show a shift from amphibian-feeding ancestors to a reptile-feeding ancestor during the mid-Cretaceous. Although this analysis is unable to model blood-feeding on dinosaurs separately from feeding on either birds or reptiles, this shift to reptile feeding may reflect the diversity of archosaurian reptiles present at the time, which includes crown lineages of many extant reptiles[42,43]. We found strong support for a mammal-feeding ancestor of *Anopheles* at 110 MYA, potentially indicating that *Anopheles* may have been feeding from early mammal lineages. Outside of the Anophelinae, our models do not find strong support for a mammal-associated ancestor for any extant group until after the K-Pg; notably, the genus *Aedes* had strong support for a mammal-associated common ancestor 62 MYA [HPD: 52–77 MYA]. As such, *Aedes* appears to have diversified alongside mammals (Fig. 2D). A similar trend is seen in feeding associations with birds, where strong support for a bird-feeding common ancestor occurs only after the K-Pg in the genera *Culex* and *Culiseta*, two genera that feed heavily upon modern birds and contain major zoonotic vectors of diseases such as West Nile virus and eastern equine encephalitis virus, both of which have reservoirs in extant birds.

Taken together, our phylogenomic results demonstrate that contemporary distributions of mosquitoes may be associated with ancient continental drift events, and that ancient associations with vertebrate hosts and flowering plants have likely shaped the evolutionary relationships of extant clades of mosquitoes. We suspect that our analyses of host associations are only the beginning of what we will learn about mosquito diversification through the study of ecological associations and mosquito phylogeny, such as in more comprehensive analyses of the evolution of larval habitat[10] or diapause[35]. Our results place the ancient origin of mosquitoes in the Triassic, during which, ancestors of extant mosquitoes likely fed on amphibians, with major diversification of genera and species in the Jurassic and Cretaceous. The diversification of mammals after the K-Pg[33] enabled the transition of numerous mosquito lineages to specialize upon these hosts, eventually giving rise to thousands of species that feed upon mammals, including species that are now human-adapted vectors of deadly pathogens, such as *Anopheles gambiae* and *Aedes aegypti*, both of which have had a profound impact on human evolution and history.

## Methods

### Taxon sampling and specimen collection
All samples were collected legally following local requirements and our research complies with all relevant ethical regulations. Insect specimens were received at NC State University's Insect Museum, pursuant to CITES permit 08US827653 (GRSCICOLL URI: http://grbio.org/cool/ij62-iybb).

Our dataset contains representatives of 268 species from nine tribes, both mosquito subfamilies, and five outgroups from three midge subfamilies (Supplementary Data 1). For most samples, data were collected via the anchored hybrid enrichment protocol described herein, but for some, previously sequenced genomes or transcriptomes were used instead (Supplementary Note 1).

For anchored hybrid enrichment, 215 mosquito species were received from collaborators, as well as samples collected by NEON (neonscience.org). All specimens were legally collected in and exported from their countries of origin. Most specimens received were whole insects; in a few cases, previous DNA extractions were received (Supplementary Data File 1 and 2). Specimens were identified to species by collaborators, and where possible, identification was confirmed by C. Sither and J. Soghigian at North Carolina State University using published keys for given regions. Specimens were stored at −20 °C until further processing.

### Anchored hybrid enrichment
We extracted genomic DNA from whole mosquito specimens using the Qiagen DNEasy kit following the manufacturer's recommendations. We quantified the DNA from each extraction using a Qubit fluorometer (High Sensitivity Kit). Isolated DNA was stored at −20 °C prior to library construction.

Library construction followed previously published anchored hybrid enrichment methods[44,45]. In short, DNA from each sample was sheared by sonication with a Covaris E2220 ultrasonicator to c. 300 bp, and we used this sheared DNA for input to a genomic DNA library preparation protocol similar to that described by Meyer and Kircher[46]. Following indexing of individual samples, we pooled samples to 48 individuals and enriched pools using the Diptera AHE kit[44], an Agilent Custom SureSelect kit (Agilent Technologies) that contains 57,681 custom-designed probes. This probe kit targets 559 loci and was constructed based on sequences from 21 total Dipteran genome and transcriptomes, including three mosquitoes: *Anopheles gambiae*, *Aedes aegypti*, and *Culex quinquefasciatus*. Libraries were sequenced either on an Illumina HiSeq 2500 (one pooled library per lane, single read mode, 100 bp—see Supplementary Data 2) or an Illumina NovaSeq 6000 (two pooled libraries per lane, paired end mode, 150 bp—see Supplementary Data 2), at the North Carolina State University Genomic Sciences Laboratory. All AHE laboratory procedures and sequencing were conducted in laboratory facilities of the North Carolina State University, Department of Entomology & Plant Pathology (Wiegmann Lab).

Demultiplexing of reads was conducted by the NCSU Genomic Sciences Laboratory. We then removed low-quality sequences and trimmed adapters using trimmomatic v 0.36[47] for samples sequenced on the HiSeq, or fastp v0.20[48] for samples sequenced on the NovaSeq. Cleaned reads were assembled using trinity v2.4[49] or SPADES[50].

### Transcriptome and genome sequence data collection
For ortholog catalog creation and to utilize the extensive genomic resources available for mosquitoes in our phylogenomic analyses, we retrieved transcriptome and genomic gene sets from GenBank or previous publications (Supplementary Data 2). A subset of these gene sets and transcriptomes were used first to create the ortholog catalog, and later used in subsequent phylogenomic analyses, described below.

## Denovo genome sequencing and assembly

We generated genome sequence data from some mosquito specimens, as described in detail, including with SOPs, in Andrade Justi et al.[51]. Briefly, e-vouchers were taken for specimens, genomic DNA was extracted using protocols developed for insect archival collections[52], libraries were constructed using KAPA HyperPlus Kits, (Roche, Pleasanton, CA) and sequenced on the NovaSeq 6000 platform at the Walter Reed Army Institute of Research. Next, we trimmed reads with Trimmomatic[47] and assembled those reads with the GATB-Minia Pipeline (https://github.com/GATB/minia) for whole genome assembly. We identified putative genes with AUGUSTUS[53] and the training species set to a mosquito (flag --species=aedes). These gene sets were used as input to Orthograph[54] with a Culicidae ortholog catalog (detailed in Supplementary Material I.5).

## Ortholog catalog creation

To integrate sequence capture data with genomic and transcriptomic data, we generated an ortholog catalog from ten mosquito genomes and transcriptomes using OMA[55,56]. Prior to our work, existing genomic resources in mosquitoes were predominantly anophelines, a subfamily that represents fewer than 15% of described mosquito species. As such, for ortholog catalog creation, we chose taxa based on available sequences that would balance taxonomic coverage of the family with quality genome sequences. As such, we used four anopheline genomes representing four subgenera, and six culicine genomes or transcriptomes from five tribes: genome sequences from *Anopheles* (*Cellia*) *gambiae* (AgamP4.11, [https://vectorbase. org/vectorbase/app/record/dataset/TMPTX_agamPimperena]), *An.* (*Anopheles*) *atroparvus* (AatrE3.1, https://vectorbase.org/vectorbase/ app/record/dataset/TMPTX_aatrEBRO]), *An.* (*Nyssorynchus*) *albimanus* (AalbS2.6, [https://vectorbase.org/vectorbase/app/record/dataset/ TMPTX_aalbSTECLA]), *An.* (*Lophophodomyia*) *squamifemur* (Asqu1.1, which we assembled early in the project), *Aedes aegypti* (AaegL5.1, [https://vectorbase.org/vectorbase/app/record/dataset/TMPTX_ aaegLVP_AGWG]) and *Aedes albopictus* (AaloF1.2, [https://vectorbase. org/vectorbase/app/record/dataset/TMPTX_aalbFoshan]) from the Aedini, *Culiseta melanura* from the Culisetini, *Culex quinquefasciatus* (CpipJ2_4, [https://vectorbase.org/vectorbase/app/record/dataset/ TMPTX_cquiJohannesburg]), and two transcriptomes from *Toxorynchites amboinensis* ([https://figshare.com/articles/dataset/Sequence_ and_functional_annotation_of_T_amboinensis_genes/1092617]) and *Tripteroides aranoides* (GGBM00000000.1, [https://www.ncbi.nlm.nih. gov/nuccore/GGBM00000000.1]) from the Toxorhynchitiini and Sabethini, respectively. Sources of genomes are listed in Supplementary Data 2.

Amino acid sequences from these genomic resources were used as input to OMA to identify orthologous gene clusters. OMA's algorithm has three phases[55,57]: (1) an all-vs-all comparison wherein each protein is aligned to all other proteins using a Smith-Waterman algorithm, (2) mutually closest putative homologs are identified based on evolutionary distances, uncertainty of inference, and gene loss, and finally, (3) orthologs are clustered into ortholog groups, where single copy ortholog groups (called OMA Groups) contain only a single amino acid sequence per species (which corresponds to a single protein-coding gene). OMA also outputs hierarchical ortholog groups that can contain paralogs, but we did not utilize these sequences in our analyses. Next, we retrieved OMA Groups that occurred in at least six of the ten genomic resources available and with at least three species per subfamily. This resulted in 7982 ortholog group alignments (OMA Groups) found in at least three species in either subfamily, with a mean number of species per OMA Group of eight.

We then retrieved the nucleotide sequences corresponding to the amino acid sequences found to be orthologous from the original gene sets and transcriptomes. Both amino acid and nucleotide sequences were used as input to the program Orthograph[54] for ortholog catalog construction following the guides available from the author of Orthograph, available on GitHub (https://github.com/mptrsen/ Orthograph). Our scripts to convert OMA output to Orthograph input are available on the Culicitree github page (https://github.com/ jsoghigian/culicitree).

## Ortholog identification and processing

Our pipeline leverages the bycatch (non-targeted regions) obtained during sequence capture to expand the potential number of orthologous genes for phylogenomic reconstruction. To ensure genes recovered from this bycatch were single copy orthologs, and to enable integration with genomic and transcriptomic data, we identified single copy orthologs in AHE assemblies, gene sets, and transcriptomes using the program Orthograph[54] with the mosquito ortholog set described above. Orthograph uses a graph-based approach to assign nucleotide or amino acid sequences to previously delineated groups of orthologous genes. Orthograph can accurately assign separate nucleotide/ amino acid sequences that match to different regions of the same ortholog, a beneficial feature in this context as different anchored hybrid enrichment probes may target different regions of the same gene. We used the default settings in Orthograph, with one exception: we filled gaps between different nucleotide sequences assigned to the same ortholog with Ns.

We also used Orthograph to identify orthologous mosquito genes specifically targeted by the anchored hybrid enrichment probeset. The original anchored hybrid enrichment probes were created prior to our creation of a Culicidae ortholog catalog and assigned different annotations corresponding to these reference genomes than those that are now available. As such, we retrieved gene sequences targeted by the AHE probes for three fly species: *Drosophila melanogaster*, *Phlebotomus papatasi*, and *Anopheles gambiae*. These three species were used in the original AHE project to identify probe sequences in sequencing reads[44]. The nucleotide sequences for these three species were passed to Orthograph, and any mosquito orthologs assigned to these gene sequences were included as ortholog gene models to be targeted by our probe sequences. Following Orthograph's assignment of putative ortholog identity to DNA fragments, we used the summarize_orthograph_results.pl script included with Orthograph to trim reference taxa from the Orthograph results, to assign nucleotide and amino acid sequences of each ortholog an identical header, and to mask stop codons for alignment purposes.

## Verification of sequence identity

As we used a mix of wild-caught females and museum specimens, we verified using BLAST[58] that all putative orthologs were fly (insect order Diptera) in origin, as opposed to genes that may have been obtained from a host (bloodmeal), or fungal, bacterial, or other contaminant. The nucleotide sequence from each ortholog was queried using blastn against a reference database of genomes from RefBase which included fly genomes, and many insect, other animal, fungal, bacterial, and protozoan genomes. Any ortholog for which the top BLAST hit was found to be not from a fly was discarded from future analyses.

## Alignment and quality control

We developed an alignment and trimming pipeline to compare different datasets with differing inclusion criteria drawn from the same set of orthologs identified by Orthograph for both nucleotide or corresponding amino acid sequences. As part of this pipeline, we developed a set of scripts to assess the presence of orthologs across the samples in our dataset, provided on the Culicitree github (https:// github.com/jsoghigian/culicitree). This allowed us to easily vary the inclusion criterion for orthologs and from which samples to align sequences. Our primary phylogenetic analyses were based on amino acid alignments of orthologs found in 75% or more of species. We also generated alignments for comparison to our primary alignments or for

additional analyses: (1) orthologs found in 90% or more of species for divergence time estimation, (2) orthologs targeted only by the AHE probes to verify probe recovery and topology, (3) and orthologs found in 75% of genomes or transcriptomes to assess the topology inferred from genomic resources alone. Results using these different datasets are presented in our Supplementary materials.

We aligned amino acid sequences from all included species per included ortholog with MAFFT G-INSI-i with 1000 iterations and the add-fragments flag[59], specifying the original ortholog catalog alignment as the base alignment, and the set of orthologs per species as the fragments to add. This strategy allowed the proper alignment of different fragments of the same gene occasionally by anchor hybrid enrichment or partial gene or transcript sequences that could be found in the incomplete genomes and transcriptomes we used. Sequences from the reference catalog were removed, and we then used trimal in gappyout (flag -gappyout) mode to trim resulting alignments, with the backtranslation option enabled and the nucleotide sequences per gene as input (the flag -backtrans). This alignment and trimming strategy kept nucleotide sequences in frame, and with the same trimming as present in amino acid sequences.

## Phylogenetic analyses

Next, we assessed outlier sequences in our individual gene alignments based on the distributions of genetic distance within the alignments across species, removing outlier sequences from alignments based on an approach similar to Tukey's 'Fences'. First, we inferred gene trees via maximum likelihood using IQ-Tree from amino acid sequences using the best-fit model chosen by IQ-Tree, and to reduce computational time, and because these trees were for screening outliers alone, we did not calculate support values and used IQ-Tree in "fast" mode[25]. We used R scripts, available on our Culicitree github (https://github.com/jsoghigian/culicitree), to assess how each taxon at each gene contributed to a particular gene tree length, by calculating the median tip-to-tip (cophenetic) distance from a given taxon to all other taxa in that gene tree, then subtracting that value from the total tip-to-tip distance for all taxa in that tree. This resulted in a relative measure of how much a given tip was contributing to the overall length of the tree. We then divided this value by the interquartile range of tip distances for that gene tree (plus a small, fixed number to account for cases with zero branch lengths) to scale this value for comparability across gene trees of different total lengths. This resulted in a branch length ratio per taxon per gene, that accounted for differences in gene tree length. Next, we defined a given taxon in a gene (evaluated as a branch length ratio) as an outlier if it was five times the interquartile range of that species, as calculated from all branch length ratios for that species. We removed such outliers from both amino acid and nucleotide alignments.

We used analysis of variance to assess whether there were significant differences in the number of orthologs in our primary dataset based on taxonomic characteristics of taxa or the data type of those sequences (AHE, transcriptome, or genome) (Supplementary Note 1).

Nucleotide saturation is known to obfuscate deep phylogenetic relationships[60], and has been previously described in other fly and mosquito datasets[9,61]. We used DAMBE v7[62] to assess saturation at each codon position in our primary dataset (Supplementary Note 2). The transition and transversion ratios and F84 distances calculated by DAMBE were exported and plotted for each codon position in R v4.1[63] using the package ggplot2[64].

Maximum likelihood analyses were performed in IQ-Tree v2.1[25]. For our primary amino acid dataset, which we discuss in the main text, we first constructed genes trees from amino acid alignments, allowing ModelFinder in IQTree to find the best model for each (flag -m MFP). We retrieved the models for each amino acid alignment, and used these in a single concatenated, partitioned analysis of all amino acid alignments. We used IQ-Tree to calculate SH-aLRT branch support

values (-alrt 1000) and ultrafast bootstrap support values (-B 1000). We also conducted three additional analyses on amino-acid datasets, the results of which we describe in the supplement: (1) on the same set of samples and orthologs as our primary dataset, but without our outlier trimming step described above and without bootstrap branch support values to reduce computational time; (2) an analysis that included all samples but only orthologs targeted by our AHE probe sequences; (3) and an analysis that included orthologs found in 75% of genomes and transcriptomes, with only genomes and transcriptomes. For these alternative analyses, we used ModelFinder on the partitioned, concatenated dataset, rather than constructing gene trees individually first as for our primary analysis. Our maximum likelihood analyses on nucleotides were primarily based on nucleotide position two due to the presence of saturation (see results, supplementary text, and Supplementary Fig. S4). We used IQ-Tree as described as above, but with ModelFinder performed on the partitioned, concatenated dataset of position two. We compared topologies for nucleotide position 2 and amino acids for our primary dataset using the R function cophyloplot from the package ape, which plots both trees and connects the same tips with dotted lines. We also performed maximum likelihood analyses on two other nucleotide datasets: an analysis where the concatenated alignment was partitioned by gene and by positions one and two within each gene but used the substitution model GTR + F to reduce computational time in estimating the model across all positions, and an analysis of the concatenated nucleotide alignment partitioned by gene and all three codon positions in which the genus-level topology was fixed to that from nucleotide position two. We did not perform analyses that included position three, as preliminary analyses prior to the sequencing of all samples indicated that the saturation at position three was misleading relationships and that the outgroup taxa were resolving within the Culicinae as sister to *Mansonia*, rendering the Culicinae subfamily non-monophyletic, which is not consistent with any previous results on the systematics of mosquitoes (see the supplement in Section III). We visualized resulting phylogenies in FigTree (available from https://github.com/rambaut/figtree), or the R package ggtree[65,66]. Topologies were rooted on the branch leading to the common ancestor of the Culicidae and the Chaoboridae, the recognized sister lineage of mosquitoes.

For coalescent-based species tree inference, we used the maximum likelihood gene trees (from above, Maximum Likelihood Analyses) as input for ASTRAL[67]. We varied the parameter -m, to assess whether including more or less complete gene trees influenced the topology resulting from ASTRAL. In addition to the default setting of including all gene trees, we also set -m to 227 (including gene trees with only 85% of taxa) and 241 (including gene trees with only 90% of taxa).

## Divergence time estimation

Our divergence time methods followed those described in dos Reis and Yang[68] for MCMCTree[29]. We used our maximum likelihood phylogeny estimated from the concatenated, partitioned amino acid alignment of all taxa as input for MCMCTree with an alignment based on orthologs found in 90% or more of species, rather than our full alignment to reduce computational requirements. We removed all outgroups, save for the Chaoboridae, from the topology for dating purposes. We kept the Chaoboridae for fossil calibration purposes.

Our fossil-based minimum-age calibrations were chosen based on the oldest fossils available that were assignable to lineages from subfamily to subgenus, based on information available on the Mosquito Taxonomic Inventory[69] and from Mosquitoes of the World[1] regarding fossil Culicidae. Following dos Reis and Yang[68], we used soft-bounded truncated Cauchy distributions for six fossil calibrations (Supplementary Table 1, Supplementary Figs. 6, 7). In addition, based on previous divergence time estimates of the order Diptera[70], we set the maximum age of our root at less than 250 million years.

We used Bayesian model selection to determine which clock model to use in MCMCTree with the R package mcmc3r (available on GitHub, https://github.com/dosreislab/mcmc3r), as clock models can have markedly different outcomes on divergence time estimates given the same fossil calibrations[71]. We used a stepping stones method suitable for large datasets[72], which uses a stationary block bootstrap method, to properly estimate Bayes factors for each of the three potential clock models MCMCTree can use: the strict clock (SC - option 1 in MCMCTree), the independent rates log-normal relaxed clock model (ILN - option 2 in MCMCTree), and the geometric Brownian motion model (GBM - option 3 in MCMCTree). We selected 32 beta values to calculate the marginal likelihood following the stepping stones method, and used these values with the MCMCTree priors, sequence alignment, phylogeny, and calibrations of our complete analysis (described above and below) for three separate 32-run analyses, one for each clock model. We used a sampling frequency of 2 and recorded 10,000 samples. We calculated the log likelihood from 100 bootstrap replicates per beta value and clock model in mcmc3r and assessed the optimal clock model based on Bayes factor values for all three models (Supplementary Note 3).

MCMCTree implements the approximate likelihood method for clock dating proposed by Thorne et al.[73] and implemented in MCMCtree[74] that reduces computation time significantly and enables the analysis of large alignments such as ours. This method calculates the gradient and Hessian matrix of the branch lengths of the topology based on the alignment and substitution model. We specified the LG+Gamma substitution model (via the aaRatefile argument in the MCMCTree control file) and ran MCMCTree and CODEML to calculate the gradient and Hessian matrix, which were then supplied along with input files to MCMCTree. Next, we used MCMCTree to sample the posterior distribution using the approximate likelihood method (usedata = 2 in the MCMCTree control file). Based on available tutorials and author recommendations, we used a uniform prior on node ages for uncalibrated nodes (BDparas = 1 1 0), and a diffuse prior on the mean substitution rate prior (2 40 1) and the rate variance parameter (sigma2_gamma = 1 10 1). Other parameters, e.g., substitution model parameters, were already set previously during approximate likelihood calculation. We set burnin to 20000, sampling frequency to 4000, and the number of samples to 10000. We chose sampling parameters as we expected they would greatly exceed the number of samples needed to properly assess if runs converged. Following recommendations of[68] for adequate effective sample size of parameters, we ran five concurrent MCMCTree runs on the same control file and evaluated convergence using Tracer and custom R scripts while MCMCTree was still running. When runs appeared to have converged (Supplementary Fig. 14) and had over 4500000 generations per run, we compared posterior means of node estimates using custom R scripts, and combined these multiple MCMC runs into a single MCMC file for analysis. We summarized the combined MCMC in MCMCTree (by setting the config file parameter print to −1) and visualized divergence time results using the R packages ggtree[65] and deeptime[75].

**Blood host database**

We conducted an extensive literature search to catalog available information on mosquito-host associations in order to assess how phylogenetic relationships among mosquitoes might be reflected in their host associations (Supplementary Note 4).

We collated a database containing nearly all mosquito blood meal studies whereby mosquitoes were sampled from field collections without the use of animal baits, and blood meals were identified by a molecular technique. The first step was to systematically obtain original research articles from the Web of Science, Google Scholar, and NCSU Summon using the following search terms and their respective combinations: 'blood', 'blood feeding', 'bloodmeal', 'blood meal',

'blood-meal', 'blood host', 'blood-host', 'Culicidae', 'feeding', 'interaction', 'mosquito', 'preference', 'vector', and 'vector-host'. Furthermore, we inspected any reference or review articles obtained in our literature search for additional articles that were not caught by our search terms or present in Web of Science, Google Scholar, or NCSU Summon. Next, we manually examined each article to determine peer-review status, study collection methods, blood meal identification methods, and collection locations. We excluded studies based on the following criteria: use of animal baits as a means for mosquito attraction, unspecified collection methodologies, unidentifiable geographic locations, and unidentifiable species samples sizes and/or study sample sizes. We extracted and recorded mosquito taxonomic information down to the nearest identifiable taxonomic unit along with study collection method(s), collection site location(s), GPS coordinates or GPS coordinate estimates, bloodmeal identification method(s), host taxonomic information down to the nearest identifiable taxonomic unit, and total blood-fed mosquitoes per blood host. Our dataset preserves the origin of each piece of data by linking each record and respective mosquito taxon to the primary data source. This allows our dataset to link both to the publication and to current mosquito taxonomic information for a given blood meal even if the original publication reported a different mosquito species epithet for an observation, i.e., our dataset respects taxonomic name changes. We used the mosquito catalog hosted by the Walter Reed Biosystematics Unit (mosquitocatalog.org) as a reference for current mosquito taxonomic information. The mosquito catalog contains nearly all recognized species names and synonyms. If a name did not match or could not be found in the mosquito catalog, we then searched through the taxonomic literature for name changes. In all cases when a mosquito name was not present in the mosquito catalog, it was due to improper Latin gender endings. There is considerable variation in how blood-meal results are reported in the literature. We attempted to grab all relevant and comparable data present within a publication without inducing systematic bias from our data collection approach. This means that some potentially relevant information is excluded, but data type comparability is preserved within the dataset. Thus, restricting our sample to all publications that test the blood contents using a molecular technique in field collected mosquitoes allows all mosquito blood host literature to be compared. In order to achieve this goal we made several explicit assumptions about the data represented in the database: (1) exclusion of mosquito double feeding events and "unknown" categories, (2) GPS coordinates can be estimated from location description information, and (3) that blood host common names can be identified to a minimum identifiable taxonomic unit.

Mosquito double feeding events have a heterogeneous record throughout the literature with some studies explicitly testing for double feeding and others ignoring these events. Moreover, prior to the ability to test for double feeding, these events likely ended up in an unknown or unspecified blood meal category if an "unknown" category was present in the paper. In addition, the "unknown" or "unspecified" categories are not always reported in a paper. Since all studies prior to 2018 only tested for vertebrate blood implicitly[38], the "unknown" category would have included possible invertebrate feeding observations, double feeding events, and blood meals that were unidentifiable due to degradation or methodological errors. Thus, collecting the "unknown" or "unspecified" data categories in blood feeding publications would contain a multitude of vague observations that would be difficult to sort out. Given this, we decided to include only data for single feeding events. Standardization of reporting across blood meal studies would greatly improve the accuracy and comparability of future syntheses.

The minimum identifiable taxonomic unit is defined here as either the identified taxon a publication reported for an organism, or if a common animal name was used, the nearest identifiable taxon for that common animal name. For instance, a 'turtle' would be reported as

order Testudines and not specified any further, while a box turtle would be *Terrapene* sp. and a common box turtle would be *Terrapene carolina* if the collection took place in the United States, since the name "box turtle" represents the genus *Terrapene* in the United States and *Terrapene carolina* represents the "common box turtle" as a common name in North Carolina and various other southern states. Fortunately, the vast majority of papers do not use common names and instead specify relevant taxonomic information. Papers that do utilize common names are rarely more specific than that of family-level identification (aside from *Homo sapiens sapiens*).

For analyses of host-associations among mosquitoes, we summarized the aforementioned host association database by species of mosquito and by class of host on which that mosquito fed. We calculated the proportion of each class a given species fed upon by dividing the number of bloodmeals from that class by the total number of bloodmeals. In our analyses, we considered this to be a measure of host association. As our literature database spanned the 20th century, numerous classifications were in place at different periods. Many early blood-meal analysis studies did not differentiate between species of vertebrate host, instead reporting only a class of host (as the assay used could typically not discriminate between related animals). This results in literature reports of, for instance, "reptile feeding." Specifically in the case of "reptile", this represented nearly a third of observations attributable to clades formerly associated as reptiles (Supplementary Table 4). As such, we chose to use the paraphyletic "reptile" in analyses to encompass Crocodilia, Squamata, and Testudines, following classifications from the early and mid 20th century. As the composition of mammals, birds, and amphibians have not changed, we did not need to make major decisions on clade composition therein.

We provide a spreadsheet that contains the references used in the database in a GitHub repository (https://github.com/jsoghigian/culicitree) and the dataset is summarized in Supplementary materials.

## A complete phylogeny of mosquitoes using taxonomy-aided complete trees

To evaluate the evolution of host associations in mosquitoes (see Phylogenetic Comparative Methods, below), we reconstructed a fully resolved phylogeny of the Culicidae using Taxonomy Aided Complete Trees[36], hereafter TACT. As we recovered host association information for some mosquitoes for which we lacked genomic data, we used our dated phylogeny and the robust taxonomic information available for mosquitoes to place unsampled tips into our phylogeny using a birth-death process, resulting in a complete sampling of all mosquitoes. We followed author's instructions for running TACT, first building a 'taxonomy tree' that generated an unresolved phylogeny based on existing taxonomy (with tact_build_taxonomic_tree and a text file of mosquito taxonomy), followed by placing unsampled lineages and tips on our molecular time tree based on that taxonomy tree (with tact_add_taxa). Two mosquito genera were not monophyletic in our analyses (see discussion in main text and below), and so we accounted for this in our input of taxonomic groups for TACT by grouping subgenera according to their previously defined associations[12,14,26,76] and following phylogenetic relationships discussed on the Mosquito Taxonomic Inventory[69], e.g., subgenera associated with the *Aedes* subgenus *Ochlerotatus* such as *Finlaya*, *Georgecraigius*, and others (Supplementary Data 3). In addition, our molecular dataset contains some subspecies, such as *Aedes notoscriptus* Red and Blue subspecies. We used only one representative per species for the backbone provided to TACT. Our taxonomy and backbone files used for TACT are available on the Culicitree Github (https://github.com/jsoghigian/culicitree).

As the birth-death process implemented in TACT is stochastic, we repeated the construction of a complete tree of the Culicidae 100 times with the same input. This resulted in 100 phylogenies that varied in position of unsampled lineages and tips. We summarized these 100 phylogenies with a maximum clade credibility tree (hereafter the TACT-MCC tree) with the R package phanghorn[77] and the function maxCladeCred, such that we could summarize additional analyses from these trees on a single topology for easier visualization, as well as to compare single-topology analyses to the set of trees.

## Phylogenetic comparative methods

We performed all phylogenetic comparative statistical analyses using R v4.1.0[63]. We summarized the mosquito blood host records from individual mosquitos to the species-level, by tallying the total number of blood meals acquired from a particular taxonomic class of hosts (i.e., amphibian, bird, mammal, or reptile—see note above regarding our usage of 'reptile' here) and calculated the proportion each class represented for each mosquito species. This yielded a dataset with 422 species, but with an additional 12 species excluded because the mosquito species could not be identified. We pruned the TACT-MCC phylogeny to include only species found in our summarized blood host dataset.

## The evolution of mosquito/blood host associations

We tested the strength of phylogenetic signal[78,79] in blood host usage—i.e., that closely related mosquito species tend to share blood hosts. We did this using a multivariate generalization[80] of Blomberg's K[79] as implemented in the function physignal from the package *geomorph* v 4.0.1[81,82]. To test whether signal strength differs significantly from zero, we randomly permuted data at tips 10,000 times for all host classes together and each host class considered separately, respectively. We further tested the strength of phylogenetic autocorrelation (an alternative measure of phylogenetic signal, reported as a Mantel test statistic[83]) for all blood host associations using a phylogenetic correlogram, as implemented in the function phyloCorrelogram from the R package *phyloSignal* v1.3[84]. We note that the Anophelinae branches early from the rest of the Culicidae and exhibits little variation in host association (i.e., almost exclusively mammals, with average mammal host association >0.92). This could lead to an erroneously strong signal for a mammalian host. Thus, we repeated the above analyses whilst excluding Anophelinae to ensure the robustness of our findings ($n = 318$) (Supplementary Note 5).

We reconstructed blood host usage over the evolutionary history of Culicidae using stochastic character mapping. We used the proportion of each host class observed for each mosquito species as a prior probability, then fit three continuous-time reversible, k-state Markov models[85–87]—equal rates, symmetrical rates, and all-rates-different—as implemented in the function make.simmap in the *phytools* package v1.0-1[88]. The equal rates model assumes a single instantaneous transition rate between all blood host classes. The symmetrical rates model allows for different transition rates between each host class, but 'forward' and 'backward' rates are equal (e.g., mammal→bird = bird→mammal). The all-rates-different model assumes different rates for all transitions. We compared these models against one another, calculating Akaike Information Criterion[89](AIC) from their likelihoods, and compared their weights[90] ($AIC_w$). As with the phylogenetic signal analysis, we repeated ancestral state reconstructions whilst excluding Anophelinae (Supplementary Note 5).

We inferred the timing of host diversification using lineage-through-time analyses as implemented in the function ltt95 in the *phytools* package[88]. We did this by downloading a posterior distribution of 100 phylogenies for each class[33,91–93] from VertLife.org. Following the taxonomy used to create each phylogeny, we pruned each posterior distribution of phylogenies to the family-level for each class, then estimated the median number of families that accumulated over time in each class.

**Reporting summary**

Further information on research design is available in the Nature Portfolio Reporting Summary linked to this article.

## Data availability

The nucleotide sequence data reported herein are archived in the NCBI, NIH SRA (Sequence Read Archive) under SRA Bioproject Number PRJNA907815 -https://www.ncbi.nlm.nih.gov/bioproject/PRJNA907815/. The phylogenetic trees generated in this study are available on our GitHub (https://github.com/jsoghigian/culicitree), https://doi.org/10.5281/zenodo.8212811. Nucleotide and amino acid alignments we analyzed, along with input files we used for analyses conducted in R, are available on Figshare, https://doi.org/10.6084/m9.figshare.23826144. We used publicly available genomes and transcriptomes for ortholog catalog creation, summarized here: *Anopheles* (*Cellia*) *gambiae* (AgamP4.11, [https://vectorbase.org/vectorbase/app/record/dataset/TMPTX_agamPimperena]), *An.* (*Anopheles*) *atroparvus* (AatrE3.1, https://vectorbase.org/vectorbase/app/record/dataset/TMPTX_aatrEBRO]), *An.* (*Nyssorynchus*) *albimanus* (AalbS2.6, [https://vectorbase.org/vectorbase/app/record/dataset/TMPTX_aalbSTECLA]), *An.* (*Lophophodomyia*) *squamifemu*r (Asqu1.1, which we assembled early in the project), *Aedes aegypti* (AaegL5.1, [https://vectorbase.org/vectorbase/app/record/dataset/TMPTX_aaegLVP_AGWG]) and *Aedes albopictus* (AaloF1.2, [https://vectorbase.org/vectorbase/app/record/dataset/TMPTX_aalbFoshan]), *Culex quinquefasciatus* (CpipJ2_4, [https://vectorbase.org/vectorbase/app/record/dataset/TMPTX_cquiJohannesburg]), and two transcriptomes from *Toxorhynchites amboinensis* ([https://figshare.com/articles/dataset/Sequence_and_functional_annotation_of_T_amboinensis_genes/1092617]) and *Tripteroides aranoides* (GGBM00000000.1,). The mosquito bloodmeal data we analyzed is provided as Supplementary Data 4. Specimen data, including collection locale or relevant publication, is available in Supplementary Data 2.

## Code availability

Scripts used in this study, such as phylogenetic and comparative methods, are available on the Culicitree Github (https://github.com/jsoghigian/culicitree), https://doi.org/10.5281/zenodo.8212811.

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

## Acknowledgements

We thank D. Mckenna and C. Mitter for comments and suggestions on an earlier version of the manuscript. Data generation and analyses were completed with help from the NCSU Genome Sciences Laboratory (GSL) and the NCSU High Performance Computing Cluster (Henry2). This work was supported by US National Science Foundation project DEB-17534376 (B.M.W., M.D.T., Y.-M.L., M.H.R., R.R.D., and G.M.), a National Institutes of Health project R01 AI 155562 (J.S., A.G.-S., and J.R.P.), and an NSERC Discovery Grant (J.S.). S.A.J. and Y.-M.L. were supported by the Global Emerging Infections Surveillance Branch of the Armed Forces Health Surveillance Division (AFHSD-GEIS) P0065_22_WR. S.A.J. is a National Research Council Research Associate at the Walter Reed Biosystematics Unit and Walter Reed Army Institute of Research. We also thank E. Johnston-Flies, M. Diallo, I. Giantsis, W. Foster, T. Sasaki, A. Faraji, and the National Ecological Observatory Network for contributing specimens. The material published reflects the views of the authors and should not be misconstrued to represent those of the U.S. Department of the Army, the U.S. Department of Defense, the National Institutes of Health, or the National Science Foundation. The funders had no role in the study design, data collection, analysis, decision to publish, or preparation of the manuscript.

## Author contributions

Conceptualization: J.S., C.S., S.A.J., M.H.R., Y.-M.L., R.C.W., M.D.T., R.R.D., and B.M.W. Methodology: J.S., C.S., G.M., M.H.R., B.K.C., and B.M.W. Investigation: J.S., C.S., B.K.C., G.M., S.A.J., M.H.R., Y.-M.L., and B.M.W. Visualization: J.S., C.S., M.H.R., and B.M.W. Funding acquisition: B.M.W., Y.-M.L., M.D.T., D.K.Y., R.R.D., and M.H.R. Specimen acquisition: J.S., C.S., Y.-M.L., R.C.W., C.J.V., T.L., S.X., A.G.-S., J.R.P., T.Z., C.H., N.D.B.-C., L.E.R., D.K.Y., M.A.S., and B.B. Project administration: B.M.W., M.H.R., and Y.-M.L.. Supervision: B.M.W., M.H.R., J.S., and Y.-M.L. Writing — original draft: J.S., R.R.D., M.H.R., C.S., and B.M.W. Writing—review and editing: J.S., M.H.R., B.M.W., S.A.J., Y.-M.L., R.C.W., T.L., M.D.T., J.R.P., R.R.D., and D.K.Y.

## Competing interests

The authors declare no competing interests.
