## [Peer Review File · Nature Communications]

REVIEWER COMMENTS

Reviewer #1 (Remarks to the Author):

In this manuscript, the authors present a phylogenetic analysis of the family Culicidae – Mosquitos. This is the first major phylogenomic study of this important fly family, with more comprehensive taxon sampling and far more loci than previous studies. The resulting well-resolved phylogeny is calibrated with fossils to estimate the timing of diversification of various clades and relate these to major geological events. The authors also compile a database of blood host records and use this to analyze evolutionary trends in hosts use. This study is a welcome addition to our understanding of Mosquito phylogenetic history and relationships and will serve as a solid foundation for future studies of these ecologically and economically important flies. The inference and analysis methods appear to be appropriate and well conducted and the results should be of broad interest. My comments are relatively few and mostly minor.

While I found the relationship between geologic events -and diversification of hosts- and the diversification of various mosquito clades intriguing, the authors might use more caution in their statements about the associations between these phenomena (e.g., p13). Although some of these inferences make sense and seem likely, they are conjectures. There may be correspondence, but this only suggests a cause or influence, it doesn't confirm it. No matter when diversification of mosquito lineages took place, the events could be ascribed to some geological or evolutionary event.

I understand that there is limited space in this paper, but I was struck by the absence of any examination or analysis of the larvae and their breeding habits/ecology, especially given that there is a wealth of information on immature stages of mosquitos. Might there be interesting evolutionary patterns of larval ecology? There is pretty much no mention of morphological traits or adaptations (of adults in the main text; there is some discussion of traits in the supplementary methods). Hopefully someone uses this phylogeny to evaluate the evolution of morphology at some point. Also, the authors state that all but two genera are monophyletic – indicating that the morphological systematists of the group resolved clades quite well without the benefits of genomic data – this might be acknowledged.

Abstract:

Isn't "difficult to discern" also "Little known"?

Major geologic events is vague

One sentence summary – replace disease organisms with disease vectors

P. 3 para1 – Are there really "so many species" that are our enemies? Apparently less than 3% vector human diseases.

P3 para 2 – species names even in this context of groups (aegypti, gambiae) are usually italicized and not capitalized, but I don't know of mosquito workers do things differently.

P3 para 3 – It is stated that we don't know how many origins of human feeding there were in Mosquitos. What does the authors phylogeny infer about this?

P5 para 2 ((species.. -> (species..

P5 para 3 – was in the early Jurassic (?), or existed in the early Jurassic, or lived...

The authors might spend a little more time explaining how there could be a 100 my discrepancy on the estimated age of the family. How could Misof et al. have been so off? Why is the current estimate more reliable? (this is discussed in the supplementary materials but it should be mentioned here).

P9 para3 – and ability to transmit...

Should blood-host be hyphenated? Sometimes it reads a bit oddly.

P10 para 2 class

(class of hosts is a bit broad, it would be interesting to look at host specificity and narrower levels)

Fig 2A. Is each point a species? Fig. 2D are the gray bands just for reference?

p. 15 para 2 – what is natural h?

P. 15 para 3 – not all name italicized

P.17 para 2 – three species are used in an ? (were used in an?)

P. 18 P. 1 topology inferred from genomic resources (genomic resources don't have a topology)

P. 18 para 3. It is unclear what the authors mean by "subtracting the tip-to-tip distances from each species' tip to tip distance"

Also later on "...as an outlier if it was..."

Which taxa were removed as outliers?

P. 18 para 4 – taxonomic characteristics is vague

P. 19 ML analyses – There is a fair amount of discussion here of the various analyses that were conducted but no real mention of how the results of the various analyses compare or what differed between them.

P. 20 – para 3 – We followed and used?

P. 21 – Para 2 – combine with previous paragraph.

P.24 para 1 – *Terrapene* should be italicized.

I read over the supplementary material and noticed several grammatical errors and awkward phrases. The authors should carefully review this material and look for such errors.

(I would not consider limitation to a single class of hosts, "striking". Actually the polyphagy of some taxa is more surprising).

Reviewer #2 (Remarks to the Author):

The reviewed manuscript with the title "An Enduring Enemy: Phylogenomics Reveals the History of Host Use in Mosquitoes" submitted by Soghigian and co-authors provides the results of phylogenetically analyzing a large number (i.e., 709) of nuclear protein-coding genes in an unprecedented number (i.e., 256) of mosquito species. The authors use the newly inferred comprehensive and robust phylogeny to trace the phylogenetic diversification of the group in time, to understand in impact of geological events (i.e., plate tectonics) on the biogeography of the group, and to shed light on the association of mosquitos with specific host groups. The study is overall impressive, solid, and – given the medical relevance of the taxon – certainly of major interest to readers of an interdisciplinary journal.

My major criticisms are:

1) Use of medians rather than confidence intervals (which are typically HUGE) when specifying and interpreting divergence time estimates. In one instance, this has led in my opinion to overinterpreting a specific result (see specific results below).

2) Too little information on what novel insights the study provides in the summary. If space is limited, summarize methods and data set size with the single term "phylogenomic". After all, large datasets and comprehensive phylogenies have become pretty standard.

3) Lack of specific sample information, such as geocoordinates, date of collection, collector. This information is particularly critical given point (4).

4) An explicit statement that all samples were legally collected in and exported from the countries of origin (see <https://www.cbd.int/abs/>).

Minor criticisms concern the language, which is at various occasions for my taste too unspecific, lax, or redundant, and inconsistencies and inaccuracies at various levels, which give the impression that the manuscript has been quickly assembled and not carefully checked prior to submission by the authors (see specific comments below). In this context, I found it very unfortunate that the manuscript does not have line numbers!

As stated above, I think that the study is overall well done and would be interesting for many readers of Nature Communications. As I expect the author to easily address my points of critique, I recommend acceptance with minor revision.

Specific comments:

Abstract: "threaten human health through the transmission of a diverse array of viruses and pathogens". It is my understanding that viruses causing human health issues are classified as pathogens (see also first sentence of the main text). Hence, "viruses and pathogens" represents a pleonasm. I suggest "pathogens, such as viruses".

"Because mosquitoes are also highly diverse" is too unspecific. Diverse in what sense? Species

richness? Morphological disparity? Ecology?

I find "709 orthologous nuclear gene sequences from 256 mosquito species" an inaccurate statement. First, orthologous genes are genes in different species that evolved from a common ancestral gene by speciation. 709 orthologous genes sequences implies that a single set of genes that diversified in mosquitos was analyzed. But this was not the case. The authors studied 709 ortholog groups, with each group containing orthologous genes. Second, I also find the term gene sequence misleading, because the authors did not study synteny (i.e., sequence of genes in genomes), but the nucleotide sequences of genes.

"origin of mosquitoes in the early Triassic (~217 mya)". Given the typically large confidence interval associated with divergence time estimates, I personally think it would be more transparent to provide the 95 % confidence limits rather than the median and mean. This applies to all divergence time estimates in the manuscript.

Please provide reference for " 400,000 deaths worldwide". It is unclear whether reference 2 and 3 refer to this statement, as they come much later and in association with a different subtopic ("Historically, ...")

"The most comprehensive analyses have focused"  better " The most comprehensive phylogenetic analyses ..."

" (e.g., the Aegypti Group (5) or the Gambiae Complex (6, 7),"  the closing bracket is missing. I also find it strange that "Aegypti Group" and "Gambiae Complex" are written with the first letters in uppercase. I would have found "Aedes aegypti group" and "Anopheles gambiae complex" a terminology more consistent with taxonomic conventions.

"evolutionary relationships"  "phylogenetic relationships" " (evolution relationships is ambiguous, as it encompasses more than just phylogenetic relationships)

"It is not yet known how many independent origins of human-feeding those 100-some species represent,". I would avoid stating any independence. If phylogenetic origins are meant, the term independence is redundant. If evolutionary origins (excl. phylogenetic origins) are meant, there is likely never and in this specific case definitively no independence, as the different lineages have shared similar ecological constraints in their history.

"at least 100 species from 11 genera"  "eleven genera"

"how specific feeding preferences may have influenced diversification and evolution in mosquitoes." What kind of diversification? Phylogenetic? Morphological? Ecological? In addition: diversification is a special case of evolution, so diversification is actually redundant.

"have shaped the contemporary diversity of mosquitoes". Again: what diversity specifically?

"A renewed understanding of mosquito phylogeny provides an explicit evolutionary context for the major transitions in mosquito feeding habits and a narrative for how these may have been influenced by the environments they inhabit and the hosts they prey upon."  Please add "for interpreting" [or some synonymous term], as an understanding does not provide an evolutionary context in the narrow (biological) sense.

"resolution of relationships among diverse lineages"  Diverse in what respect?  "phylogenetically diverse"

"to obtain and sequence hundreds of orthologous genes" My understanding is that the authors studied orthologs of hundreds of genes in fresh and museum specimens. The nucleotide sequences of a gene in different species are orthologs and hence orthologous. But the different genes in a given species are not orthologous. This statement is therefore misleading.

"This phylogenomic dataset is the largest yet assembled for mosquitoes,"  "This phylogenomic

dataset is the largest yet assembled for phylogenetically studying mosquitoes,"

"with species from both subfamilies"  "with species from both currently recognized subfamilies"

"we recovered 709 orthologous gene sequences found in more than 203 (>75%) of the species sampled"  "we recovered the orthologous nucleotide sequences of 709 single-copy genes in more than 203 (>75%) of the species sampled"

"Our taxon sampling (number of species) more than doubles previous phylogenetic studies of mosquitoes and samples 40 times as many loci."  "Our taxon sampling (number of species) more than doubles that of previous phylogenetic studies of mosquitoes and samples 40 times as many loci as these studies did".

"To account for saturation (see supplemental material)"  As them manuscript has been submitted to an interdisciplinary journal, many readers will possibly not know the meaning of saturation in this context. I suggest to write instead: "To account for rapid phylogenetic signal decay at the nucleotide sequence level (see supplemental material)"

"we analyzed alignments of amino acids and the second nucleotide position in codons."  "we analyzed multiple sequence alignments of amino acids and of second codon position nucleotides, respectively."

" We inferred maximum likelihood phylogenies in IQTree 2 (25) for both nucleotide position two and amino acid alignments."  "We inferred phylogenies using the maximum likelihood optimality criterion implemented in IQTree 2 (25) and analyzing the amino acid and nucleotide sequence alignments, respectively."

"evolutionary relationships"  "phylogenetic relationships" (evolution relationships is ambiguous, as it encompasses more than just phylogenetic relationships)

"support for the monophyly of both existing mosquito subfamilies"  "support for the monophyly of both current recognized mosquito subfamilies" ("existing subfamilies" implies that there are also non-existing subfamilies)

"Within the subfamily Culicinae, the species-poor tribe Aedeomyiini (a pantropical tribe of only seven species) is the earliest diverging lineage, followed by the Uranotaeniini."  The sister lineage has exactly the same age. Better phrase: "Within the subfamily Culicinae, the species-poor Aedeomyiini (a pantropical tribe of only seven species) is the earliest diverging tribe, followed by the Uranotaeniini."

"and until the early 2000s (26) the genus was separated as a subfamily due to its unique behavior and morphology."  "and until the early 2000s (26) the genus was placed in a monotypic subfamily due to the unique behavior and morphology of its species."

"consistent with recent analyses with more limited data"  "consistent with recent studies that analyzed less data"

"Those two genera are the diverse and medically"  Superfluous whitespace after diverse. Diverse in what regard? Species richness? Morphology? Ecology?

"((species of which"  Superfluous opening bracket.

"these diverse clades" Again: In what regard diverse? Species richness?

" Bayesian divergence time analyses"  "Bayesian divergence time estimations" (for the sake of variety)

"Triassic (~ 217 MYA, HPD: 188–250 MYA),"  "Triassic (~ 217 MYA [HPD: 188–250 MYA])," (for the sake of consistency)

"the age of mosquitoes"  " the phylogenetic age of mosquitoes"

" This age precedes any existing mosquito fossil,"  "This age precedes that of any existing mosquito fossil,"

"Our analyses indicate that the last common ancestors of the two extant subfamilies of Culicidae, the Culicinae and the Anophelinae, was the early Jurassic, near the Toarcian Warm Interval (~ 179 MYA [HPD:147–213 MYA] (Figure 1 – B), 100 MY older than suggested by Misof et al. (22) in their seminal phylogenomic study on the evolutionary history of insects."  "Our analyses indicate that the last common ancestors of the two extant subfamilies of Culicidae, the Culicinae and the Anophelinae, lived during early Jurassic, near the Toarcian Warm Interval (~ 179 MYA [HPD:147–213 MYA] (Figure 1 – B), and was thus 100 MY older than suggested by Misof et al. (22) in their seminal phylogenomic study on the evolutionary history of insects."

"The earliest diverging lineages of the Culicinae,"  the sister lineage have the same age. Better: "The earliest diverging tribes of the Culicinae,"

"This time coincides with the diversification of flowering plants (31),"  "This time coincides with the time during which the diversification of flowering plants took place (31),"

"Fig. 1. Evolutionary relationships"  "Fig. 1. Phylogenetic relationships" (evolution encompasses more than phylogeny)

"Culicidae), as inferred by maximum likelihood and dated using a fossil-calibrated, relaxed clock analysis"  "Culicidae), as inferred when applying the maximum likelihood optimality criterion and dated using a fossil-calibrated relaxed clock analysis"

"The analysis contains 256 species, with two Chaoborid outgroups, and amino acid sequences from 709 genes with a total amino acid alignment length of 525,000 sites."  "The analysis is based on the analysis of amino acid sequences of 709 genes (525,000 aligned sites) from a total 256 species). Chaoborid were used as outgroups for rooting of the phylogeny."

"Americas, whose divergence times correlate with major geological events"  "Americas, whose divergence times correlate with the dates of major geological events".

"These ancient divergences continue"  "These resulting ancient divergences continue" (the preceding sentence does to mention any divergences)

"mosquito diversity" What diversity specifically? Species diversity? Ecological diversity?

"This date coincides with the formation of the channel separating South America and Africa in the equatorial zone"  " This date coincides with the estimated time at which the channel separating South America and Africa in the equatorial zone formed"

"distributions indicate ancient isolation"  "distributions indicate ancient geographical isolation"

"This age is slightly younger than the period of deepening of the central and southern South Atlantic that occurred between 100 and 85 MYA"  "The divergence thus happened after the deepening of the central and southern South Atlantic that occurred 85–100 MYA". Note that the confidence interval overlaps with the time interval of period, so it cannot be ruled out that the two events were causally connected.

"87 MYA [HPD:70-106]"  "87 MYA [HPD:70–106]"

"consistent with continental drift events influencing the diversity of many present-day lineages."  "consistent with the idea of continental drift events having influenced the

diversity of many present-day lineages."

"a rapid diversification in both clades took place 40-50 MYA."  "a rapid lineage diversification in both clades took place 40–50 MYA."

"albopictus, diverged 31 MYA"  "albopictus, diverged 31 MYA from each other"

"evaluation of how evolutionary relationships"  "evaluation of how phylogenetic relationships"

"We combined our phylogenomic analysis with a database"  "We linked insights from our phylogenomic analysis with data on contemporary blood host information"

"Mosquito-host associations are determined by ... from ELISA to PCR."  please provide reference substantiating this statement.

"is known to specialize on fish"  "is known to be specialized on fish"

"particular Class of host"  "particular class of host"

"0.37%), and a pair of unlucky amphibians"  "0.37%) and amphibians"

"These individual-level preferences": It is my understanding that a preference for something can be inferred from choice tests, as these allow for controlling potential confounding factors, and the authors do not present such tests. The observed differences could, for example, be explained by abundance differences of host lineages in different habitats. Hence, the term "preference" should be avoided.

"evolutionary relationships among Culicidae are"  "phylogenetic relationships among Culicidae are"

"Fig. 2. Macroevolutionary analyses of blood-feeding preference"  "Fig. 2. Macroevolutionary analyses of host usage" (see comment above of the term "preference")

"Vertebrate host preference contains"  "Vertebrate host usage contains"

"Among the Culicinae, closely related species have more similar blood-hosts, while more distant lineages differ in blood-host preference, ..."  "Among the Culicinae, closely related species tend to feed on blood of closely related hosts, whereas distantly related species tend to feed on blood of distantly related hosts"

"reconstruction on blood-host preference"  "reconstruction on blood-host usage"

"While our analysis of divergence times"  "While our estimation of divergence times"

"While our analysis of divergence times in mosquitoes highlights how ancient mosquito lineages reflect continental drift events,"  "While our divergence time estimates in mosquitoes highlights the possible role of continental drift events for the divergence of ancient mosquito lineages,"

"demonstrating the robustness of these findings"  "demonstrating the robustness of this finding" (the reconstructions are the sample points that support the idea (singular) of an amphibian-feeding ancestor).

"Although this analysis is unable to model blood-feeding on dinosaurs separately from either birds or reptiles"  "Although this analysis is unable to model blood-feeding on dinosaurs separately from feeding blood of either birds or reptiles"

"and that ancient associations with vertebrate hosts and flowering plants shaped extant clades." Shaped in what regard?

"Present-day blood host associations have both significant phylogenetic signal, and apparent association with the expansion of contemporary hosts, as seen in a strong association in the timing of diversification between *Culex* and *Aedes* mosquitoes." I am having difficulties understanding the second part of the sentence. What is meant with "expansion of contemporary hosts"? The sentence includes the word "association" three times.

"Our results place the ancient origin of mosquitoes in the Triassic, where ancestors of mosquitoes likely fed on amphibians "  "Our results place the evolutionary origin of mosquitoes in the Triassic, during which ancestors of mosquitoes likely fed on amphibians"

"The diversification of mammal feeding clades after the K-Pg boundary allowed for species such as *An. gambiae* and *Ae. aegypti* to eventually become human-adapted vectors of deadly pathogens, with a subsequent profound impact on human evolution and history."  While the sentence sounds great and nicely closes the story arc, I do not see what novel aspect it provides. The fact that *An. gambiae* and *Ae. aegypti* exist and feed on human blood means that conditions in the past allowed them to evolve and become human adapted. And because we are talking about two lineages, diversification was a prerequisite for the two species to exist. I am fully aware that the authors meant with "diversification" that of all mammal-feeding clades, but it cannot be concluded that diversification in two more than two lineages were necessary for *An. gambiae* and *Ae. aegypti* to evolve. And because diversification is not quantified in this statement, it would even be valid if only two extant mammal-feeding species (i.e., *An. gambiae* and *Ae. aegypti*) existed (given that they diverged from each other after the K-Pg boundary). Thus, the final statement of the study simply trivial and I strongly suggest to close this otherwise impressive investigation with a more inspiring insight.

Methods

Taxon Sampling and Specimen Collection.

I am missing a statement that all analyzed genetic samples have been legally collected in the countries of origin. I am also missing specific information on the analyzed samples, such as country of origin, geocoordinates, voucher specimens, date of collection, collector. The latter two data are particularly important for legal reasons.

"-20C"  "-20 °C" (multiple times)

Anchored Hybrid Enrichment

"protocol similar to Meyer and Kircher (50)."  " protocol similar to that given by Meyer and Kircher (50)."

Ortholog Catalog Creation

"*Anopheles* (*Cellia*) *gambiae*"  The parentheses should not be in italics.

"*An.* (*Anopheles*) *atroparvus*"  Genus, subgenus and specific epithet must be in italics.

"*An.* (*Nyssorynchus*) *albimanus*"  The parentheses should not be in italics.

"*An.* (*Lophophodomyia*) *squamifemur*"  The parentheses should not be in italics

Peptide sequences  Amino acid sequences

"orthologous gene clusters"  "ortholog gene clusters"

"orthologs are clustered into orthologous groups"  "orthologs are clustered into ortholog groups"

"single copy orthologous groups"  "single-copy ortholog groups"

"only a single gene sequence"  " only a single amino acid sequence (corresponding to single protein-coding gene)"

"OMA also outputs hierarchical orthologous groups"  "OMA also outputs hierarchical ortholog groups"

"7982 orthologous gene alignments"  "7982 ortholog group alignments"

Ortholog Identification and Processing

"obtained during sequence capture"  "obtained during nucleotide sequence capture"

" These three species are used in and original AHE project"  "These three species are used in the original AHE project"?

" The gene sequences for these three species"  " The nucleotide sequences for these three species"

Verification of Sequence Identity

"using BLAST"  please provide reference

"Each ortholog sequence"  What kind of sequence? Nucleotide or amino acid. What search algorithm (e.g., tblastx)?

Alignment and Phylogenetic Analysis

"which samples to align sequences from"  " which samples to align nucleotide or corresponding amino acid sequences from"

"based on alignments of orthologs"  " based on amino acid alignments of orthologs"

"tip-to-tip genetic distance"  genetic distance in ambiguous. What distance measure was applied on what data?

"First, we calculated gene trees using IQ-Tree from amino acid sequences"  What substitution model? Also mention optimality criterion, even if indirectly perhaps clear.

"We used R scripts"  Please provide link to download R scripts.

"This resulted in a branch length ratio per taxa per gene"  " This resulted in a branch length ratio per taxon per gene"

"We used R scripts to assess the tip-to-tip genetic distance for each taxon at each gene, and the contribution of that taxon to the overall length of that gene tree, by subtracting the tip-to-tip distance from each species' tip to tip distance, then dividing this value by the interquartile range of tip distances (plus a small, fixed number to account for cases with zero branch lengths)."  I am having trouble understanding what was done. First, a tip-to-tip distance involves two taxa: how can a single taxon have a single tip-to-tip distance? Second: what is the difference between the first and the second tip-to-tip distance in the statement "by subtracting the tip-to-tip distance from each species tip to tip distance": Third: what tip distances are we talking about in the statement "then dividing this value by the interquartile range of tip distances". You mean tip-to-tip distances, correct? And which ones do you mean? All (across all possible species pairs) or those that includes a given species?

"as an outlier when if it was five"  either "when" or "if"

"obfuscate deep evolutionary relationships"  "obfuscate deep phylogenetic relationships"

" datasets(64, 9)."  "datasets (64, 9)."

Please use a consistent style to specify versions: "DAMBE 7", "R v4.1", "IQ-Tree version 2.1"

Maximum Likelihood Analyses

The title is confusing. First, the preceding chapter is called Alignment and Phylogenetic Analysis.

The chapter Maximum Likelihood Analyses covers a major fraction of the phylogenetic analyses, so the preceding chapter is obviously mislabeled. Second, maximum likelihood is a generic method to estimate parameters of some probability distribution. This is the reason why I suggested to also write in the main text that a phylogenetic estimate was obtained by applying the maximum likelihood optimality criterion. I suggest to name the chapter Phylogenetic Analysis of Supermatrices, rather than Maximum Likelihood Analyses.

"We retrieved the models for each gene,"  the previous sentence called the entities alignments. It would be good to consistently use either gene or alignment.

Coalescence-like ASTRAL Analysis  Instead of naming the analysis by the software that implemented it, what about using "Summary-Coalescent Species Tree Inference"?

Please name the software consistently: "ASTRAL", "Astral"

"Clock Model Evaluation"  Shouldn't this be better a subchapter of Divergence Time Estimation?

Approximate Likelihood Calculation in MCMCTree.  Shouldn't this be better a subchapter of Divergence Time Estimation? And shouldn't it be better stated in the header WHAT is calculated rather than HOW (i.e., "Approximate Likelihood Calculation" and "in MCMCTree")?

"ggtree 69)"  "ggtree (69)"

Bloodhost Database

"to assess how evolutionary relationships"  "to assess how phylogenetic relationships"

Please provide at the end of the chapter a link to the database that lists all references and extracted information.

Data Descriptors and Assumptions

This chapter should be included in the chapter Bloodhost Database, as it specifies the entries in the database.

"In all cases when a mosquito name was not present in the mosquito catalog was due to improper Latin gender endings."  " In all cases when a mosquito name was not present in the mosquito catalog, it was due to improper Latin gender endings."

" A wide array of reporting styles exists"  the term "style" is too unspecific. It is not clear to me what specifically is meant.

A Complete Phylogeny of Mosquitoes using Taxonomy-Aided Complete Trees.

"analyses to the set of trees ."  superfluous whitespace before dot

"R ver. 4.1.0"  please use a consistent style to specify the versions of software (see above)

" from a particular hosts' taxonomic class"  " from a particular taxonomic class of hosts"

"from a particular hosts' taxonomic class (i.e., mammal, bird, reptile, or amphibian)"  "reptile" is not a class, but an outdated name for a para- (extant lineages) or polyphyletic (extant and fossil lineages) group of lineages. Please apply a contemporary terminology!

Phylogenetic signal

The header is too generic. I suggest "Phylogenetic signal for blood host association"

"strength of phylogenetic signal (83, 84) in blood host preference"  " strength of phylogenetic signal (83, 84) in blood host association"

"to share blood host preferences"  "to share a similar blood host association"

"generalization(85)"  missing whitespace  "generalization (85)"

"geomorph ver. 4.0.1(86, 87)."  missing whitespace  "geomorph ver. 4.0.1 (86, 87)."

"for all host classes together and each host class considered separately"  "for all host classes together and for each host class separately, respectively"

"test statistic(88)"  missing whitespace  "test statistic (88)"

"little variation in host preference"  "little variation in blood host association"

"for mammalian host preference"  "for mammalian blood host association"

Ancestral State Reconstruction

Header too generic. Reconstruction of what trait? I suggestion "Tracing the evolution of blood host associations".

"We reconstructed blood host preference"  "We reconstructed blood host associations"

" We used the proportion of each host class observed for each species"  " We used the proportion of each host class observed for each mosquito species"

"(90 - 92)"  "(90-92)"

" The symmetrical rates model assumes different transition rates between each host class,"  "The symmetrical rates model allows different transition rates between each host class," (It is my understanding [but please double-check] that the model does not demand the transition rates to be different!)

"(33, 45 - 47)"  "(33, 45-47)"

RESPONSE TO REVIEWERS' COMMENTS

Reviewer #1 (Remarks to the Author):

>In this manuscript, the authors present a phylogenetic analysis of the family Culicidae – Mosquitos. This is the first major phylogenomic study of this important fly family, with more comprehensive taxon sampling and far more loci than previous studies. The resulting well-resolved phylogeny is calibrated with fossils to estimate the timing of diversification of various clades and relate these to major geological events. The authors also compile a database of blood host records and use this to analyze evolutionary trends in hosts use. This study is a welcome addition to our understanding of Mosquito phylogenetic history and relationships and will serve as a solid foundation for future studies of these ecologically and economically important flies. The inference and analysis methods appear to be appropriate and well conducted and the results should be of broad interest. My comments are relatively few and mostly minor.

We thank this reviewer for their thorough review, and for their complimentary thoughts on our study. We have attempted to address this reviewer's feedback thoroughly, which we describe below. Owing to the detailed and important feedback of this reviewer we feel our manuscript has been substantially improved, and so we thank this review for their time and effort in reviewing our manuscript.

>While I found the relationship between geologic events -and diversification of hosts- and the diversification of various mosquito clades intriguing, the authors might use more caution in their statements about the associations between these phenomena (e.g., p13). Although some of these inferences make sense and seem likely, they are conjectures. There may be correspondence, but this only suggests a cause or influence, it doesn't confirm it. No matter when diversification of mosquito lineages took place, the events could be ascribed to some geological or evolutionary event.

We appreciate that we could emphasize with greater degree that our statements are conjecture, and that they should be interpreted with a degree of caution. To that end, we have made sure to cushion many of our statements on association between divergences and continental geologic events. For instance, we introduce our discussion on how the breakup of Gondwana appears associated with mosquito distributions with this statement: "We find an intriguing association between mosquito lineages and major geologic events, such that the breakup of Gondwana during the Cretaceous may have shaped the distribution of extant mosquito lineages." And, throughout our discussion of the breakup of Gondwana and its association with anopheline distributions, our statements now attempt to make clear that these patterns may reflect association, rather than definitively do. In addition, on pg13 (Page 12, Line 184 in the edited version of our manuscript), we now say "While our estimates of divergence times in mosquitoes highlights how ancient mosquito lineages **may** reflect continental drift events...". We have made several such alterations throughout our manuscript to reflect that these are intriguing correlations, rather than definitive tests.

>I understand that there is limited space in this paper, but I was struck by the absence of any examination or analysis of the larvae and their breeding habits/ecology, especially given that there is a

wealth of information on immature stages of mosquitos. Might there be interesting evolutionary patterns of larval ecology? There is pretty much no mention of morphological traits or adaptations (of adults in the main text; there is some discussion of traits in the supplementary methods). Hopefully someone uses this phylogeny to evaluate the evolution of morphology at some point. Also, the authors state that all but two genera are monophyletic – indicating that the morphological systematists of the group resolved clades quite well without the benefits of genomic data – this might be acknowledged.

We wholeheartedly agree with this reviewer that there are certainly interesting patterns in the evolution larval ecology and morphology! Within the largest tribe of mosquitoes, the Aedini, this has already been examined (Soghigian et al. 2017, BMC Evolutionary Biology), but we recognize that this study lacks the sampling of ours. However, it was outside the scope of this study to also build a database on larval habitat across the family, as although such data do exist for many species, our initial efforts had limitations on what we could collate and analyze. We do feel that this is certainly a direction that we (or other researchers) should pursue, and so we acknowledge this in our last paragraph: “...We suspect that our analyses on bloodhost associations are only the beginning of what we will learn about mosquito diversification through the study of ecological associations and mosquito phylogeny, such as more comprehensive analyses of the evolution of larval habitat (11) or diapause (35)...”

In addition, we do agree that it is impressive how well morphological systematics have performed at delineating mosquito clades. We agree with this reviewer that this should be acknowledged, and now modify the last sentence of our discussion on the two genera that are not monophyletic to include: “It is clear that these clades have complex histories that will continue to require taxonomic study and clarification, but this highlights that overall, the morphological systematics of mosquitoes has performed quite well at delineating clades, if not always in determining the associations between them (see our supplemental information for further discussions on mosquito systematics in light of these results).”

>Abstract:

>Isn't “difficult to discern” also “Little known”?

True! Here, we were trying to convey both the difficulty the field has had with properly reconstructing relationships due to limitations of previous datasets AND the generally poor sampling of most mosquitoes. We have edited this sentence to simply read “The phylogeny of mosquitoes has remained poorly characterized due to difficulty in taxonomic sampling and limited availability of genomic data beyond the most important vector species.” As

>Major geologic events is vague

We've changed this sentence to refer to continental drift specifically.

>One sentence summary – replace disease organisms with disease vectors

We appreciate the reviewer catching this typo and have made the change they suggested.

>P. 3 para1 – Are there really “so many species” that are our enemies? Apparently less than 3% vector human diseases.

We feel that this conservative estimate is still many species, particularly compared to other major vector or parasite groups, where there are typically far fewer species involved. We recognize that this is in reference to a later sentence in which we state 100-some species are involved in disease transmission to humans. However, since our submission, a new paper highlights that an additional 200 species may be involved in disease transmission (Yee et al. 2022), and so we have updated this sentence to read: "Out of roughly 3600 mosquito species globally, approximately 100 species from eleven genera certainly play a role in the transmission of disease to humans (1) and another 200 are likely or potential vector species (Yee et al. 2022)."

>P3 para 2 – species names even in this context of groups (*aegypti*, *gambiae*) are usually italicized and not capitalized, but I don't know of mosquito workers do things differently.

Among mosquito systematists, recognized unranked species groups are not italicized (the *Aegypti* Group, <https://mosquito-taxonomic-inventory.myspecies.info/simpletaxonomy/term/8724>). However, we have modified this line such that *Gambiae* Complex has been replaced by the more common "*Anopheles gambiae* complex."

>P3 para 3 – It is stated that we don't know how many origins of human feeding there were in Mosquitos. What does the authors phylogeny infer about this?

This is a good question, but it is difficult to address quantitatively with the data we have. Given where humanity evolved and when, it seems highly likely this number is quite close to the total number of species that feed from humans (exceptions may exist among African *Anopheles* species).

>P5 para 2 ((species.. -> (species..

Thank you for this catch! We've removed one parenthesis.

>P5 para 3 – was in the early Jurassic (?), or existed in the early Jurassic, or lived...

As per the reviewer's suggestion, we have added the word "in". Thanks!

>The authors might spend a little more time explaining how there could be a 100 my discrepancy on the estimated age of the family. How could Misof et al. have been so off? Why is the current estimate more reliable? (this is discussed in the supplementary materials but it should be mentioned here).

We appreciate this reviewer's point here, and we agree that it is worth including a bit more information here. We have added this sentence to the end of the paragraph that starts on L140 on Page 5: "The robustness of our sampling and our integration of multiple mosquito fossils, along with analytical differences (see Methods and supplemental text), likely account for the differences in our estimates and theirs, as Misof et al. used only one fossil calibration across all flies and had only two mosquito species in their analysis."

>P9 para3 – and ability to transmit...

Thank you for catching another missing word.

>Should blood-host be hyphenated? Sometimes it reads a bit oddly.

In the literature, it is sometimes hyphenated, and sometimes not. .

>P10 para 2 class

>(class of hosts is a bit broad, it would be interesting to look at host specificity and narrower levels)

We agree with this reviewer that that is a particularly exciting future direction for this research. At present, we are unable to conduct such analyses, as though

>Fig 2A. Is each point a species? Fig. 2D are the gray bands just for reference?

We have updated our captions to reflect that in Fig 2 A, individual points are species, and gray bands define geologic time periods.

>p. 15 para 2 – what is natural h?

This was a typo.

>P. 15 para 3 – not all name italicized

All species names are now italicized.

>P.17 para 2 – three species are used in an ? (were used in an?)

This part has been changed to “These three species were used in the...”

>P. 18 P. 1 topology inferred from genomic resources (genomic resources don't have a topology)

Changed to “...topology inferred from genomic resources...”

>P. 18 para 3. It is unclear what the authors mean by “subtracting the tip-to-tip distances from each species' tip to tip distance”

>Also later on “...as an outlier if it was...”

>Which taxa were removed as outliers?

We have substantially re-written this section of the manuscript to clarify our methods of outlier removal. For each gene tree, we evaluated how much a taxon in that gene tree contributed to the overall length of the tree, and we removed (from the alignment used to generate the gene tree) those taxa whose contribution to the total length of a tree was an outlier relative to its typical contribution. We provide additional information in our supplement, and the scripts we used to detect these outliers will be available on our GitHub. The relevant, clarified section of our methods reads:

“Next, we assessed outlier sequences in our individual gene alignments based on the distributions of genetic distance within the alignments across species, removing outlier sequences from alignments based on an approach similar to Tukey's ‘Fences’. First, we inferred gene trees via maximum likelihood using IQ-Tree from amino acid sequences using the best-fit model chosen by IQ-Tree, and to reduce computational time, and because these trees were for screening outliers alone, we did not calculate support values and used IQ-Tree in “fast” mode (25). We used R scripts, available on our Culicitree github (<https://github.com/jsoghigian/culicitree>), to assess how each taxon at each gene contributed to a particular gene tree length, by calculating the median tip-to-tip (cophenetic) distance from a given taxon to all other taxa in that gene tree, then subtracting that value from the total tip-to-tip distance for all taxa in that tree. This resulted in a relative measure of how much a given tip was

contributing to the overall length of the tree. We then divided this value by the interquartile range of tip distances for that gene tree (plus a small, fixed number to account for cases with zero branch lengths) to scale this value for comparability across gene trees of different total lengths. This resulted in a branch length ratio per taxon per gene, that accounted for differences in gene tree length. Next, we defined a given taxon in a gene (evaluated as a branch length ratio) as an outlier if it was five times the interquartile range of that species, as calculated from all branch length ratios for that species. We removed such outliers from both amino acid and nucleotide alignments.”

>P. 18 para 4 – taxonomic characteristics is vague

This refers to ortholog recovery at the subfamily, genus, or tribe level. We have clarified that our presentation of these results is in the supplement.

>P. 19 ML analyses – There is a fair amount of discussion here of the various analyses that were conducted but no real mention of how the results of the various analyses compare or what differed between them.

We have clarified throughout the methods where specific sections relate to the supplement. For instance, in this paragraph, we now say “For our primary amino acid dataset, **which we discuss in the main text,**...” and “We also conducted three additional analyses on amino-acid datasets, **the results of which we describe in the supplement...**”

>P. 20 – para 3 – We followed and used?

This was a repeat of reference 71, dos Reis and Yang 2019, that was omitted from the text unintentionally.

>P. 21 – Para 2 – combine with previous paragraph.

Done.

>P.24 para 1 – Terrapene should be italicized.

Done.

>I read over the supplementary material and noticed several grammatical errors and awkward phrases. The authors should carefully review this material and look for such errors.

>(I would not consider limitation to a single class of hosts, “striking”. Actually the polyphagy of some taxa is more surprising).

We have proofread our supplemental materials and also edited them for content, including our reference to striking host associations.

Reviewer #2 (Remarks to the Author):

>The reviewed manuscript with the title "An Enduring Enemy: Phylogenomics Reveals the History of Host Use in Mosquitoes" submitted by Soghigian and co-authors provides the results of phylogenetically analyzing a large number (i.e., 709) of nuclear protein-coding genes in an unprecedented number (i.e., 256) of mosquito species. The authors use the newly inferred comprehensive and robust phylogeny to trace the phylogenetic diversification of the group in time, to understand in impact of geological events (i.e., plate tectonics) on the biogeography of the group, and to shed light on the association of mosquitos with specific host groups. The study is overall impressive, solid, and — given the medical relevance of the taxon — certainly of major interest to readers of an interdisciplinary journal.

We are grateful to the reviewer for their kind comments, interest in our analyses, and for their thorough and comprehensive review of our manuscript. We have carefully considered the reviewer's feedback and believe that, through incorporating their suggestions, we have improved our manuscript significantly.

My major criticisms are:

>1) Use of medians rather than confidence intervals (which are typically HUGE) when specifying and interpreting divergence time estimates. In one instance, this has led in my opinion to overinterpreting a specific result (see specific results below).

We now are careful to report HPDs in all circumstances in which we also report a node estimate in our text. We also have, throughout the text, added statements which we hope help reduce any perception of overinterpretation of results.

>2) Too little information on what novel insights the study provides in the summary. If space is limited, summarize methods and data set size with the single term "phylogenomic". After all, large datasets and comprehensive phylogenies have become pretty standard.

We have attempted to highlight a few of the novel insights from our study better in the abstract:

"Mosquitoes have profoundly affected human history and continue to threaten human health through the transmission of a diverse array of pathogens. The phylogeny of mosquitoes has remained poorly characterized due to difficulty in taxonomic sampling and limited availability of genomic data beyond the most important vector species. Here, we used phylogenomic analysis of 709 single copy ortholog groups from 256 mosquito species to produce a strongly supported phylogeny that resolves the position of the major disease vector species and the major mosquito lineages. Our analyses support an origin of mosquitoes in the early Triassic (217 MYA [HPD: 188–250 MYA]), **considerably older than previous estimates**. Moreover, **we utilize an extensive database of host associations for mosquitoes to show that mosquitoes have shifted to feeding upon the blood of mammals numerous times**, and that mosquito diversification and host-use patterns within major lineages appear to coincide in earth history both with major continental drift events and with the diversification of vertebrate classes."

>3) Lack of specific sample information, such as geocoordinates, date of collection, collector. This information is particularly critical given point (4).

This version of the manuscript includes Supplemental Table 2, which contains information on the origin of all samples / sequences used in the study, including location of collection and date.

>4) An explicit statement that all samples were legally collected in and exported from the countries of origin (see <https://www.cbd.int/abs/>).

We have added such a statement to our methods: "All specimens were legally collected in and exported from their countries of origin."

>Minor criticisms concern the language, which is at various occasions for my taste too unspecific, lax, or redundant, and inconsistencies and inaccuracies at various levels, which give the impression that the manuscript has been quickly assembled and not carefully checked prior to submission by the authors (see specific comments below). In this context, I found it very unfortunate that the manuscript does not have line numbers!

We hope we have addressed this reviewer's concerns regarding our language use. We apologize for the lack of line numbers and hope that, should this reviewer review our manuscript again, the line numbers are shown.

>As stated above, I think that the study is overall well done and would be interesting for many readers of Nature Communications. As I expect the author to easily address my points of critique, I recommend acceptance with minor revision.

Again, we thank the reviewer for their comments and hope we have adequately addressed their concerns.

>Specific comments:

>Abstract: "threaten human health through the transmission of a diverse array of viruses and pathogens". It is my understanding that viruses causing human health issues are classified as pathogens (see also first sentence of the main text). Hence, "viruses and pathogens" represents a pleonasm. I suggest "pathogens, such as viruses".

This reviewer is indeed correct regarding viruses. We have re-written the abstract slightly, and to save space, have simplified this statement by removing viruses entirely.

>"Because mosquitoes are also highly diverse" is too unspecific. Diverse in what sense? Species richness? Morphological disparity? Ecology?

We have changed the opening of this sentence to "Due to their high species diversity and near-cosmopolitan distribution, the mosquito phylogeny..."

>I find "709 orthologous nuclear gene sequences from 256 mosquito species" an inaccurate statement. First, orthologous genes are genes in different species that evolved from a common ancestral gene by speciation. 709 orthologous genes sequences implies that a single set of

genes that diversified in mosquitos was analyzed. But this was not the case. The authors studied 709 ortholog groups, with each group containing orthologous genes. Second, I also find the term gene sequence misleading, because the authors did not study synteny (i.e., sequence of genes in genomes), but the nucleotide sequences of genes.

We have updated the text to reflect the reviewer's concerns regarding the accuracy of the term orthologous nuclear gene sequences to instead read "Here, we used phylogenomic analysis of 709 single copy ortholog groups from 256 mosquito species..."

>"origin of mosquitoes in the early Triassic (~217 mya)". Given the typically large confidence interval associated with divergence time estimates, I personally think it would be more transparent to provide the 95 % confidence limits rather than the median and mean. This applies to all divergence time estimates in the manuscript.

We have updated our divergence time estimates to always include HPDs. We agree this is important for transparency!

>Please provide reference for " 400,000 deaths worldwide". It is unclear whether reference 2 and 3 refer to this statement, as they come much later and in association with a different subtopic ("Historically, ...")

Done.

>"The most comprehensive analyses have focused"  better " The most comprehensive phylogenetic analyses ..."

Done.

>" (e.g., the Aegypti Group (5) or the Gambiae Complex (6, 7),"  the closing bracket is missing. I also find it strange that "Aegypti Group" and "Gambiae Complex" are written with the first letters in uppercase. I would have found "Aedes aegypti group" and "Anopheles gambiae complex" a terminology more consistent with taxonomic conventions.

Naming conventions in mosquitoes do differ from some other insect groups and, as far as these terms go, they are used conventionally in this form (although *Anopheles gambiae* complex is also in wide usage). While the Gambiae Complex is used, we agree that *Anopheles gambiae* complex is in wider usage and so have altered the text as per the reviewer's recommendations. However, the Aegypti Group is an unranked species group in the subgenus *Stegomyia* (<https://mosquito-taxonomic-inventory.myspecies.info/simpletaxonomy/term/8724>) and so we maintain the usage here.

>"evolutionary relationships"  "phylogenetic relationships" " (evolution relationships is ambiguous, as it encompasses more than just phylogenetic relationships)

Done.

>"It is not yet known how many independent origins of human-feeding those 100-some species represent,". I would avoid stating any independence. If phylogenetic origins are meant, the term independence is redundant. If evolutionary origins (excl. phylogenetic origins) are meant, there

>is likely never and in this specific case definitively no independence, as the different lineages have shared similar ecological constraints in their history.

As here we were referring to phylogenetic origins, we have removed independent.

>"at least 100 species from 11 genera"  "eleven genera"

Done.

>"how specific feeding preferences may have influenced diversification and evolution in mosquitoes." What kind of diversification? Phylogenetic? Morphological? Ecological? In addition: diversification is a special case of evolution, so diversification is actually redundant.

To address the redundancy highlighted by this reviewer, we've removed the clause "and evolution" from this sentence.

>"have shaped the contemporary diversity of mosquitoes". Again: what diversity specifically?

Changed "diversity" to "species richness".

>"A renewed understanding of mosquito phylogeny provides an explicit evolutionary context for the major transitions in mosquito feeding habits and a narrative for how these may have been influenced by the environments they inhabit and the hosts they prey upon."  Please add "for interpreting" [or some synonymous term], as an understanding does not provide an evolutionary context in the narrow (biological) sense.

Following the reviewer's suggestion, we have added "for interpreting" to this sentence, such that it now reads "A renewed understanding of mosquito phylogeny provides an explicit evolutionary context for interpreting the major transitions in mosquito feeding habits and a narrative for how these may have been influenced by the environments they inhabit and the hosts they prey upon."

>"resolution of relationships among diverse lineages"  Diverse in what respect?  "phylogenetically diverse"

Done.

>"to obtain and sequence hundreds of orthologous genes" My understanding is that the authors studied orthologs of hundreds of genes in fresh and museum specimens. The nucleotide sequences of a gene in different species are orthologous and hence orthologous. But the different genes in a given species are not orthologous. This statement is therefore misleading.

Sequencing is conducted on a large scale across multiple specimens from different species (described further in methods). We hope that clarifies this statement.

>"This phylogenomic dataset is the largest yet assembled for mosquitoes,"  "This phylogenomic dataset is the largest yet assembled for phylogenetically studying mosquitoes,"

Done.

>"with species from both subfamilies"  "with species from both currently recognized subfamilies"

Done.

>"we recovered 709 orthologous gene sequences found in more than 203 (>75%) of the species sampled"  "we recovered the orthologous nucleotide sequences of 709 single-copy genes in more than 203 (>75%) of the species sampled"

We appreciate this reviewer's suggestion to increase the clarity and specificity of our language and have made this change.

>"Our taxon sampling (number of species) more than doubles previous phylogenetic studies of mosquitoes and samples 40 times as many loci."  "Our taxon sampling (number of species) more than doubles that of previous phylogenetic studies of mosquitoes and samples 40 times as many loci as these studies did".

Done.

>"To account for saturation (see supplemental material)"  As them manuscript has been submitted to an interdisciplinary journal, many readers will possibly not know the meaning of saturation in this context. I suggest to write instead: "To account for rapid phylogenetic signal decay at the nucleotide sequence level (see supplemental material)"

We appreciate this suggestion and, following the review's suggestion, have re-written this sentence: "To account for rapid phylogenetic signal decay at the nucleotide sequence level in certain codon positions (saturation - see supplemental material)"

>"we analyzed alignments of amino acids and the second nucleotide position in codons."  "we analyzed multiple sequence alignments of amino acids and of second codon position nucleotides, respectively."

Done.

>" We inferred maximum likelihood phylogenies in IQTree 2 (25) for both nucleotide position two and amino acid alignments."  "We inferred phylogenies using the maximum likelihood optimality criterion implemented in IQTree 2 (25) and analyzing the amin acid and nucleotide sequence alignments, respectively."

Done.

>"evolutionary relationships"  "phylogenetic relationships" (evolution relationships is ambiguous, as it encompasses more than just phylogenetic relationships)

Done.

>"support for the monophyly of both existing mosquito subfamilies"  "support for the monophyly of both current recognized mosquito subfamilies" ("existing subfamilies" implies that there are also non-existing subfamilies)

Done.

>"Within the subfamily Culicinae, the species-poor tribe Aedeomyiini (a pantropical tribe of only seven species) is the earliest diverging lineage, followed by the Uranotaeniini."  The sister lineage has exactly the same age. Better phrase: "Within the subfamily Culicinae, the species-poor Aedeomyiini (a pantropical tribe of only seven species) is the earliest diverging tribe, followed by the Uranotaeniini."

Done.

>"and until the early 2000s (26) the genus was separated as a subfamily due to its unique behavior and morphology."  "and until the early 2000s (26) the genus was placed in a monotypic subfamily due to the unique behavior and morphology of its species."

Done.

>"consistent with recent analyses with more limited data"  "consistent with recent studies that analyzed less data"

Done.

>"Those two genera are the diverse and medically"  Superfluous whitespace after diverse. Diverse in what regard? Species richness? Morphology? Ecology?

White space removed and "diverse" replaced with "species-rich".

>"(species of which"  Superfluous opening bracket.

Removed the extra parenthesis.

>"these diverse clades" Again: In what regard diverse? Species richness?

Removed "diverse", as reiterating the diversity of these groups is redundant.

>" Bayesian divergence time analyses"  "Bayesian divergence time estimations" (for the sake of variety)

Done.

>"Triassic (~ 217 MYA, HPD: 188–250 MYA),"  "Triassic (~ 217 MYA [HPD: 188–250 MYA])," (for the sake of consistency)

Done.

>"the age of mosquitoes"  " the phylogenetic age of mosquitoes"

Done.

>" This age precedes any existing mosquito fossil,"  "This age precedes that of any existing mosquito fossil,"

Done.

>"Our analyses indicate that the last common ancestors of the two extant subfamilies of Culicidae, the Culicinae and the Anophelinae, was the early Jurassic, near the Toarcian Warm Interval (~ 179 MYA [HPD:147–213 MYA] (Figure 1 – B), 100 MY older than suggested by Misof et al. (22) in their seminal phylogenomic study on the evolutionary history of insects."  "Our analyses indicate that the last common ancestors of the two extant subfamilies of Culicidae, the Culicinae and the Anophelinae, lived during early Jurassic, near the Toarcian Warm Interval (~ 179 MYA [HPD:147–213 MYA] (Figure 1 – B), and was thus 100 MY older than suggested by Misof et al. (22) in their seminal phylogenomic study on the evolutionary history of insects."

We have adjusted this sentence to include the clause "lived during."

>"The earliest diverging lineages of the Culicinae,"  the sister lineage have the same age. Better: "The earliest diverging tribes of the Culicinae,"

Done.

>"This time coincides with the diversification of flowering plants (31),"  "This time coincides with the time during which the diversification of flowering plants took place (31),"

We have decided to keep our current wording as we feel that, while the reviewer's suggestion is certainly accurate, our statement is clear in context.

>"Fig. 1. Evolutionary relationships"  "Fig. 1. Phylogenetic relationships" (evolution encompasses more than phylogeny)

Done.

>"Culicidae), as inferred by maximum likelihood and dated using a fossil-calibrated, relaxed clock analysis"  "Culicidae), as inferred when applying the maximum likelihood optimality criterion and dated using a fossil-calibrated relaxed clock analysis"

We understand the suggestion, but prefer to retain our original description as specifically referring to maximum likelihood as an optimality criterion is not necessary in this case.

>"The analysis contains 256 species, with two Chaoborid outgroups, and amino acid sequences from 709 genes with a total amino acid alignment length of 525,000 sites."  "The analysis is based on the analysis of amino acid sequences of 709 genes (525,000 aligned sites) from a total 256 species). Chaoborid were used as outgroups for rooting of the phylogeny."

Changed to something similar—"The analysis is based on the analysis of amino acid sequences of 709 genes (525,000 aligned sites) from 256 species. Two Chaoborid outgroups were used to root the phylogeny".

>"Americas, whose divergence times correlate with major geological events"  "Americas, whose divergence times correlate with the dates of major geological events".

Done.

>"These ancient divergences continue"  "These resulting ancient divergences continue" (the preceding sentence does to mention any divergences)

Done.

>"mosquito diversity" What diversity specifically? Species diversity? Ecological diversity?

Changed to "mosquito species diversity".

>"This date coincides with the formation of the channel separating South America and Africa in the equatorial zone"  " This date coincides with the estimated time at which the channel separating South America and Africa in the equatorial zone formed"

Changed to "...coincides with the estimated time of formation...."

>"distributions indicate ancient isolation"  "distributions indicate ancient geographical isolation"

Done.

>"This age is slightly younger than the period of deepening of the central and southern South Atlantic that occurred between 100 and 85 MYA"  "The divergence thus happened after the deepening of the central and southern South Atlantic that occurred 85–100 MYA". Note that the confidence interval overlaps with the time interval of period, so it cannot be ruled out that the two events were causally connected.

Done.

>"87 MYA [HPD:70-106]"  "87 MYA [HPD:70–106]"

Done.

>"consistent with continental drift events influencing the diversity of many present-day lineages."  "consistent with the idea of continental drift events having influenced the diversity of many present-day lineages."

Done.

>"a rapid diversification in both clades took place 40-50 MYA."  "a rapid lineage diversification in both clades took place 40–50 MYA."

Done.

>"albopictus, diverged 31 MYA"  albopictus, diverged 31 MYA from each other"

Changed to "...diverged from one another..."

>"evaluation of how evolutionary relationships"  "evaluation of how phylogenetic relationships"

We have left this wording as we refer broadly here and not only to our analyses.

>"We combined our phylogenomic analysis with a database"  "We linked insights from our phylogenomic analysis with data on contemporary blood host information"

Done.

>"Mosquito-host associations are determined by ... from ELISA to PCR."  please provide reference substantiating this statement.

We now clarify, in the next sentence, that our references and database are available on GitHub and DataDryad.

>"is known to specialize on fish"  "is known to be specialized on fish"

Done.

>"particular Class of host"  "particular class of host"

Done.

>"0.37%), and a pair of unlucky amphibians"  "0.37%) and amphibians"

We prefer to maintain the wording as it emphasizes the relative rarity of amphibian observations for this taxon.

>"These individual-level preferences": It is my understanding that a preference for something can be inferred from choice tests, as these allow for controlling potential confounding factors, and the authors do not present such tests. The observed differences could, for example, be explained by abundance differences of host lineages in different habitats. Hence, the term "preference" should be avoided.

This is a valid point and we thank the reviewer for bringing it to our attention. Following the reviewer's suggestions below, "preference" has been changed to "association" or "usage" throughout the manuscript.

>"evolutionary relationships among Culicidae are"  "phylogenetic relationships among Culicidae are"

Done.

>"Fig. 2. Macroevolutionary analyses of blood-feeding preference"  "Fig. 2. Macroevolutionary analyses of host usage" (see comment above of the term "preference")

We have edited Fig 2 and the caption to no longer refer to preference, but rather, association.

>"Vertebrate host preference contains"  "Vertebrate host usage contains"

Done.

>"Among the Culicinae, closely related species have more similar blood-hosts, while more distant lineages differ in blood-host preference, ..."  "Among the Culicinae, closely related species tend to feed on blood of closely related hosts, whereas distantly related species tend to feed on blood of distantly related hosts"

We've opted to simply delete "preference".

>"reconstruction on blood-host preference"  "reconstruction on blood-host usage"

Done.

>"While our analysis of divergence times"  "While our estimation of divergence times"

Done.

>"While our analysis of divergence times in mosquitoes highlights how ancient mosquito lineages reflect continental drift events,"  "While our divergence time estimates in mosquitoes highlights the possible role of continental drift events for the divergence of ancient mosquito lineages,"

Done.

>"demonstrating the robustness of these findings"  "demonstrating the robustness of this finding" (the reconstructions are the sample points that support the idea (singular) of an amphibian-feeding ancestor).

Done.

>"Although this analysis is unable to model blood-feeding on dinosaurs separately from either birds or reptiles"  "Although this analysis is unable to model blood-feeding on dinosaurs separately from feeding blood of either birds or reptiles"

We have adjusted the wording of this sentence to read "...separately from feeding on either birds or reptiles.."

>"and that ancient associations with vertebrate hosts and flowering plants shaped extant clades." Shaped in what regard?

Added "...shaped the evolutionary relationships of..." to clarify that we mean evolutionary relationships.

>"Present-day blood host associations have both significant phylogenetic signal, and apparent association with the expansion of contemporary hosts, as seen in a strong association in the timing of diversification between Culex and Aedes mosquitoes." I am having difficulties

understanding the second part of the sentence. What is meant with "expansion of contemporary hosts"? The sentence includes the word "association" three times.

We've changed the wording here to be more direct: "Present-day blood host usage have both significant phylogenetic signal, and strong association in the timing of diversification between *Culex* and *Aedes* mosquitoes and the timing of diversification of contemporary host lineages.

>"Our results place the ancient origin of mosquitoes in the Triassic, where ancestors of mosquitoes likely fed on amphibians "  "Our results place the evolutionary origin of mosquitoes in the Triassic, during which ancestors of mosquitoes likely fed on amphibians"

Done.

>"The diversification of mammal feeding clades after the K-Pg boundary allowed for species such as *An. gambiae* and *Ae. aegypti* to eventually become human-adapted vectors of deadly pathogens, with a subsequent profound impact on human evolution and history."  While the sentence sounds great and nicely closes the story arc, I do not see what novel aspect it provides. The fact that *An. gambiae* and *Ae. aegypti* exist and feed on human blood means that conditions in the past allowed them to evolve and become human adapted. And because we are talking about two lineages, diversification was a prerequisite for the two species to exist. I am fully aware that the authors meant with "diversification" that of all mammal-feeding clades, but it cannot be concluded that diversification in two more than two lineages were necessary for *An. gambiae* and *Ae. aegypti* to evolve. And because diversification is not quantified in this statement, it would even be valid if only two extant mammal-feeding species (i.e., *An. gambiae* and *Ae. aegypti*) existed (given that they diverged from each other after the K-Pg boundary). Thus, the final statement of the study simply trivial and I strongly suggest to close this otherwise impressive investigation with a more inspiring insight.

We appreciate that our previous closing sentence did not provide particularly novel insight. We have re-written our closing paragraph, and we hope that, as a whole, it highlights the novelty of our work and provides potential future directions that might inspire future researchers to study the phylogeny of mosquitoes.

Methods

Taxon Sampling and Specimen Collection.

>I am missing a statement that all analyzed genetic samples have been legally collected in the countries of origin. I am also missing specific information on the analyzed samples, such as country of origin, geocoordinates, voucher specimens, date of collection, collector. The latter two data are particularly important for legal reasons.

We have added the following statement to the methods: "All specimens were legally collected in and exported from their countries of origin." In addition, Supplemental Table 2 contains information on origin of specimens, country of origin, locale, and collector, whenever such information is available – we also provide references to publications or BioProjects from

which sequence data was drawn.

>"-20C"  "-20 °C" (multiple times)

Done

Anchored Hybrid Enrichment

>"protocol similar to Meyer and Kircher (50)."  " protocol similar to that given by Meyer and Kircher (50)."

Changed to "...to that described by..."

Ortholog Catalog Creation

>"Anopheles (Cellia) gambiae"  The parentheses should not be in italics.

>"An. (Anopheles) atroparvus"  Genus, subgenus and specific epithet must be in italics.

>"An. (Nyssorynchus) albimanus"  The parentheses should not be in italics.

>"An. (Lophophodomyia) squamifemur"  The parentheses should not be in italics

All italics fixed.

>Peptide sequences  Amin acid sequences

>"orthologous gene clusters"  "ortholog gene clusters"

>"orthologs are clustered into orthologous groups"  "orthologs are clustered into ortholog groups"

>"single copy orthologous groups"  "single-copy ortholog groups"

>"only a single gene sequence"  " only a single amino acid sequence (corresponding to single protein-coding gene)"

>"OMA also outputs hierarchical orthologous groups"  "OMA also outputs hierarchical ortholog groups"

>"7982 orthologous gene alignments"  "7982 ortholog group alignments"

We have made all changes as suggested, save for minor differences in wording.

Ortholog Identification and Processing

>"obtained during sequence capture"  "obtained during nucleotide sequence capture"

As the method is colloquially referred to as "sequence capture", although it always captures nucleotide sequences, we have left this wording.

>" These three species are used in and original AHE project"  "These three species are used in the original AHE project"?

>" The gene sequences for these three species"  " The nucleotide sequences for these three species"

We made both of these corrections, thank you.

Verification of Sequence Identity

>"using BLAST"  please provide reference

Done

>"Each ortholog sequence"  What kind of sequence? Nucleotide or amino acid. What search algorithm (e.g., tblastx)?

This sentence has been re-written, and now begins: "The nucleotide sequence from each ortholog was queried using blastn against a reference database of genomes..."

Alignment and Phylogenetic Analysis

>"which samples to align sequences from"  " which samples to align

Changed to "from which samples to align sequences".

>nucleotide or corresponding amino acid sequences from"

Done

>"based on alignments of orthologs"  " based on amino acid alignments of orthologs"

Done

>"tip-to-tip genetic distance"  genetic distance in ambiguous. What distance measure was applied on what data?

We have added "cophenetic" to clarify the distance we calculated.

>"First, we calculated gene trees using IQ-Tree from amino acid sequences"  What substitution model? Also mention optimality criterion, even if indirectly perhaps clear.

Substitution models were determined by IQ-Tree's modelfinder, per default behavior when not specifying a model. We have also clarified these were phylogenies inferred by maximum likelihood.

>"We used R scripts"  Please provide link to download R scripts.

We have clarified these scripts will be available on our GitHub.

>"This resulted in a branch length ratio per taxa per gene"  " This resulted in a branch length ratio per taxon per gene"

Done.

>"We used R scripts to assess the tip-to-tip genetic distance for each taxon at each gene, and the contribution of that taxon to the overall length of that gene tree, by subtracting the tip-to-tip distance from each species' tip to tip distance, then dividing this value by the interquartile range of tip distances (plus a small, fixed number to account for cases with zero branch lengths)."  I am having trouble understanding what was done. First, a tip-to-tip distance involves two taxa: how can a single taxon have a single tip-to-tip distance? Second: what is the difference between the first and the second tip-to-tip distance in the statement "by subtracting the tip-to-tip distance from each species tip to tip distance": Third: what tip distances are we talking about in the statement "then dividing this value by the interquartile range of tip distances". You mean tip-to-tip distances, correct? And which ones do you mean? All (across all possible species pairs) or those that includes a given species?

We have significantly re-written this section: "We used R scripts, available on our *CulicITree* github (<https://github.com/jsoghigian/culicITree>), to assess how each taxon at each gene contributed to a particular gene tree length, by calculating the median tip-to-tip (cophenetic) distance from a given taxon to all other taxa in that gene tree, then subtracting that value from the total tip-to-tip distance for all taxa in that tree. This resulted in a relative measure of how much a given tip was contributing to the overall length of the tree. We then divided this value by the interquartile range of tip distances for that gene tree (plus a small, fixed number to account for cases with zero branch lengths) to scale this value for comparability across gene trees of different total lengths. This resulted in a branch length ratio per taxon per gene, that accounted for differences in gene tree length. Next, we defined a given taxon in a gene (evaluated as a branch length ratio) as an outlier if it was five times the interquartile range of that given species. We removed such outliers from both amino acid and nucleotide alignments."

We hope this has clarified our method of outlier sequences within gene alignments. Essentially, this method identifies branches that are unusually long in the alignment, relative to how long sequences are in that alignment, and how long sequences for that particular taxon normally are relative to all other taxa in the alignment.

>"as an outlier when if it was five"  either "when" or "if"

Changed to "if"

>"obfuscate deep evolutionary relationships"  "obfuscate deep phylogenetic relationships"

Done.

>" datasets(64, 9)."  "datasets (64, 9)."

Done.

>Pleased use a consistent style to specify versions: "DAMBE 7", "R v4.1", "IQ-Tree version 2.1"

We have unified our style of referring to software version numbers.

Maximum Likelihood Analyses

>The title is confusing. First, the preceding chapter is called Alignment and Phylogenetic Analysis. The chapter Maximum Likelihood Analyses covers a major fraction of the phylogenetic analyses, so the preceding chapter is obviously mislabeled. Second, maximum likelihood is a generic method to estimate parameters of some probability distribution. This is the reason why I suggested to also write in the main text that a phylogenetic estimate was obtained by applying the maximum likelihood optimality criterion. I suggest to name the chapter Phylogenetic Analysis of Supermatrices, rather than Maximum Likelihood Analyses.

We can understand and apologize for the confusion in our section headers. We have changed the name of the previous section to "Alignment and Quality Control", which better encompasses the methods we present there. Then, we renamed this section to "Phylogenetic Analyses of Supermatrices."

>"We retrieved the models for each gene,"  the previous sentence called the entities alignments. It would be good to consistently use either gene or alignment.

Here, our choice of gene was to differentiate from the concatenated amino acid alignment. However, the reviewer makes a good point on consistency and so we have rewritten this sentence as: "We retrieved the models for each amino acid alignment, and used these in a single concatenated, partitioned analysis of all amino acid alignments." In addition, we have added "concatenated" to a few instances of "alignment" to clarify we are not referring to individual gene alignments.

>Coalescence-like ASTRAL Analysis  Instead of naming the analysis by the software that implemented it, what about using "Summary-Coalescent Species Tree Inference"?

Done.

>Please name the software consistently: "ASTRAL", "Astral"

Changed all instances to "ASTRAL"

>"Clock Model Evaluation"  Shouldn't this be better a subchapter of Divergence Time Estimation?

>Approximate Likelihood Calculation in MCMCTree.  Shouldn't this be better a subchapter of Divergence Time Estimation? And shouldn't it be better stated in the header WHAT is calculated rather than HOW (i.e., "Approximate Likelihood Calculation" and "in MCMCTree")?

We have placed both of these sections under the heading Divergence Time Estimation.

>"ggtree 69)"  "ggtree (69)"

Done.

Bloodhost Database

>"to assess how evolutionary relationships"  "to assess how phylogenetic relationships"

Done

>Please provide at the end of the chapter a link to the database that lists all references and extracted information.

Our intention is to release the database on our GitHub, along with various code and scripts. We have made this clear at the end of this section (it will also be in our Data Availability statement): "We provide a spreadsheet that contains the references used in the database creation, along with the host associations we analyzed, on our GitHub repository (<https://github.com/jsoghigian/culicitre>)." We have clarified this in our Data Availability statement, as well.

Field Code Changed

Data Descriptors and Assumptions

>This chapter should be included in the chapter Bloodhost Database, as it specifies the entries in the database.

Agreed – this header has been removed.

>"In all cases when a mosquito name was not present in the mosquito catalog was due to improper Latin gender endings."  " In all cases when a mosquito name was not present in the mosquito catalog, it was due to improper Latin gender endings."

Done.

>" A wide array of reporting styles exists"  the term "style" is too unspecific. It is not clear to me what specifically is meant.

Here, we are referring to how variable results reporting is for blood-meal analyses – we have rewritten this sentence as follows: "There is considerable variation in how blood-meal results are reported in the literature."

>A Complete Phylogeny of Mosquitoes using Taxonomy-Aided Complete Trees.

>"analyses to the set of trees ."  superfluous whitespace before dot

Done.

>"R ver. 4.1.0"  please use a consistent style to specify the versions of software (see above)

Done

>" from a particular hosts' taxonomic class"  " from a particular taxonomic class of hosts"

Done.

>"from a particular hosts' taxonomic class (i.e., mammal, bird, reptile, or amphibian)"  "reptile" is not a class, but an outdated name for a para- (extant lineages) or polyphyletic (extant and fossil lineages) group of lineages. Please apply a contemporary terminology!

We have clarified that we used the paraphyletic group of reptiles owed to the range of ages of studies and the lack, at times, of detailed taxonomic information on the blood-host. Almost a third of all observations were reported only as "reptile." We now address this in the methods:

"As our literature database spanned the 20th century, numerous classification schemes were in place at different periods. Many early blood-meal analysis studies did not differentiate between particular species of vertebrate host, instead reporting only a class of host (as the assay used could typically not discriminate between related animals). This resulted in literature reports only of, for instance, "reptile feeding." Specifically in the case of "reptile", this represented nearly a third of observations attributable to clades formerly associated as reptiles (see Supplemental Table 7). As such, we chose to use the paraphyletic "reptile" in analyses to encompass Crocodylia, Squamata, and Testudines, following classification schemes in the early and mid 20th century. As the composition of mammals, birds, and amphibians have not changed in that time, we did not need to make decisions on clade composition therein."

Phylogenetic signal

>The header is too generic. I suggest "Phylogenetic signal for blood host association"

Changed to Phylogenetic signal of blood host usage.

>"strength of phylogenetic signal (83, 84) in blood host preference"  " strength of phylogenetic signal (83, 84) in blood host association"

Changed to "blood host usage"

>"to share blood host preferences"  "to share a similar blood host association"

We opted to remove preferences and change to "blood hosts".

>"generalization(85)"  missing whitespace  "generalization (85)"

Done.

>"geomorph ver. 4.0.1(86, 87)."  missing whitespace  "geomorph ver. 4.0.1 (86, 87)."

Done.

>"for all host classes together and each host class considered separately"  "for all host classes together and for each host class separately, respectively"

Done

>"test statistic(88))"  missing whitespace  "test statistic (88))"

Done.

>"little variation in host preference"  "little variation in blood host association"

Changed "preference" to "usage".

>"for mammalian host preference"  "for mammalian blood host association"

Changed to "...a mammalian host."

Ancestral State Reconstruction

>Header too generic. Reconstruction of what trait? I suggestion "Tracing the evolution of blood host associations".

Done.

>"We reconstructed blood host preference"  "We reconstructed blood host associations"

Changed to "blood host usage".

>" We used the proportion of each host class observed for each species"  " We used the proportion of each host class observed for each mosquito species"

Done.

>"(90 - 92)"  "(90-92)"

Done.

>" The symmetrical rates model assumes different transition rates between each host class,"  "The symmetrical rates model allows different transition rates between each host class," (It is my understanding [but please double-check] that the model does not demand the transition rates to be different!)

Yes, symmetrical rates allows for different rates—rather than assuming that they are different. We've changed the phrasing as suggested by the reviewer.

>"(33, 45 - 47)"  "(33, 45-47)"

Done.

REVIEWERS' COMMENTS

Reviewer #1 (Remarks to the Author):

The authors have done a very nice job of responding to the reviewer comments and suggestions, and the manuscript is much improved. I have looked over the author responses to reviewer comments and perused the main text and I pretty much have no further suggestions. The manuscript is clear, interesting, and engaging and it represents a significant step (leap?) forward in our understanding of the evolution of this very important group of flies.

A couple tiny issues that I noticed were:

One-Sentence Summary: although strictly the term coincidentally is appropriate, it's connotation is that there is no relationship between these things. Either delete this term or change to coincident with or concurrently with, etc.

L. 323 analyses of host associations

Reviewer #2 (Remarks to the Author):

The authors have responded to all my comments and addressed almost all of my concerns, and in particular all of my major points. I found only some very few minor issues in the revised version of the manuscript text:

line 166: MA  MYA, and missing MYA after 87

line 167: This time coincides with the diversification of flowering plants  This time coincides with that the diversification of flowering plants

line 211: 52-77 MYA  52-77 MYA

What I find awkward is that the term "host preference" is still used in the manuscript in statements where it appear not really justified to me because the semantic ambiguity of the term. I found this especially confusing, because the authors stated that in their response letter that they replaced the term throughout the manuscript by the terms "host association" and "host usage". I would personally not have used the term preference in context of this study, but I acknowledge that this is also a matter of writing style. I wanted to mention it nonetheless in case that the replacement was accidentally incomplete.

In conclusion, I congratulate the authors for this impressive study, and I recommend the editor publication of the manuscript.

RESPONSE TO REVIEWERS' COMMENTS

Reviewer #1 (Remarks to the Author):

The authors have done a very nice job of responding to the reviewer comments and suggestions, and the manuscript is much improved. I have looked over the author responses to reviewer comments and perused the main text and I pretty much have no further suggestions. The manuscript is clear, interesting, and engaging and it represents a significant step (leap?) forward in our understanding of the evolution of this very important group of flies.

A couple tiny issues that I noticed were:

One-Sentence Summary: although strictly the term coincidentally is appropriate, it's connotation is that there is no relationship between these things. Either delete this term or change to coincident with or concurrently with, etc.

We removed the one sentence summary from the final version of the manuscript and have checked to make sure that 'coincidentally' is not used in the context elsewhere in the manuscript.

L. 323 analyses of host associations

corrected 'on' to 'of' as suggested.

Reviewer #2 (Remarks to the Author):

The authors have responded to all my comments and addressed almost all of my concerns, and in particular all of my major points. I found only some very few minor issues in the revised version of the manuscript text:

line 166: MA  MYA, and missing MYA after 87

Corrected, now line 171

line 167: This time coincides with the diversification of flowering plants  This time coincides with that the diversification of flowering plants

Corrected, now line 172

line 211: 52-77 MYA  52–77 MYA

Corrected, now line 203

What I find awkward is that the term "host preference" is still used in the manuscript in statements where it appear not really justified to me because the semantic ambiguity of the term. I found this especially confusing, because the authors stated that in their response letter that they replaced the term throughout the manuscript by the terms "host association" and "host usage". I would personally not have used the term preference in context of this study, but I acknowledge that this is also a matter of writing style. I wanted to mention it nonetheless in case that the replacement was accidentally incomplete.

In conclusion, I congratulate the authors for this impressive study, and I recommend the editor publication of the manuscript.

Thank you, we have checked the full manuscript and can confirm that the term host preference has been modified or deleted in each case.

Brian Wiegmann, NCSU, Aug 31, 2023